# Borda Regret Minimization for Generalized Linear Dueling Bandits

## Abstract

Dueling bandits are widely used to model preferential feedback prevalent in many applications such as recommendation systems and ranking. In this paper, we study the Borda regret minimization problem for dueling bandits, which aims to identify the item with the highest Borda score while minimizing the cumulative regret. We propose a rich class of generalized linear dueling bandit models, which cover many existing models. We first prove a regret lower bound of order $\Omega(d^{2/3}T^{2/3})$ for the Borda regret minimization problem, where $d$ is the dimension of contextual vectors and $T$ is the time horizon. To attain this lower bound, we propose an explore-then-commit type algorithm for the stochastic setting, which has a nearly matching regret upper bound $\widetilde{O}(d^{2/3}T^{2/3})$. We also propose an EXP3-type algorithm for the adversarial linear setting, where the underlying model parameter can change in each round. Our algorithm achieves an $\widetilde{O}(d^{2/3}T^{2/3})$ regret, which is also optimal. Empirical evaluations on both synthetic data and a simulated real-world environment are conducted to corroborate our theoretical analysis.

## 1 Introduction

Multi-armed bandits (MAB) (Lattimore & Szepesvári, 2020) is an interactive game where in each round, an agent chooses an arm to pull and receives a noisy reward as feedback. In contrast to numerical feedback considered in classic MAB settings, preferential feedback is more natural in various online learning tasks including information retrieval Yue & Joachims (2009), recommendation systems Sui & Burdick (2014), ranking Minka et al. (2018), crowdsourcing Chen et al. (2013), etc. Moreover, numerical feedback is also more difficult to gauge and prone to errors in many real-world applications. For example, when provided with items to shop or movies to watch, it is more natural for a customer to pick a preferred one than scoring the options. This motivates *Dueling Bandits* (Yue & Joachims, 2009), where the agent repeatedly pulls two arms at a time and is provided with feedback being the binary outcome of "duels" between the two arms.

In dueling bandits problems, the outcome of duels is commonly modeled as Bernoulli random variables due to their binary nature. In each round, suppose the agent chooses to compare arm $i$ and $j$, then the binary feedback is assumed to be sampled independently from a Bernoulli distribution. For a dueling bandits instance with $K$ arms, the probabilistic model of the instance can be fully characterized by a $K \times K$ preference probability matrix with each entry being: $p_{i,j} = \mathbb{P}(\text{arm } i \text{ is chosen over arm } j)$.

In a broader range of applications such as ranking, "arms" are often referred to as "items". We will use these two terms interchangeably in the rest of this paper. One central goal of dueling bandits is to devise a strategy to identify the "optimal" item as quickly as possible, measured by either sample complexity or cumulative regret. However, the notion of optimality for dueling bandits is way harder to define than for multi-armed bandits. The latter can simply define the arm with the highest numerical feedback as the optimal arm, while for dueling bandits there is no obvious definition solely dependent on $\{p_{i,j} | i, j \in [K]\}$.

The first few works on dueling bandits imposed strong assumptions on $p_{i,j}$. For example, Yue et al. (2012) assumed that there exists a true ranking that is coherent among all items, and the preference probabilities must satisfy both strong stochastic transitivity (SST) and stochastic triangle inequality (STI). While relaxations like weak stochastic transitivity (Falahatgar et al., 2018) or relaxed stochas-

tic transitivity (Yue & Joachims, 2011) exist, they typically still assume the true ranking exists and the preference probabilities are consistent, i.e., $p_{i,j} > \frac{1}{2}$ if and only if $i$ is ranked higher than $j$. In reality, the existence of such coherent ranking aligned with item preferences is rarely the case. For example, $p_{i,j}$ may be interpreted as the probability of one basketball team $i$ beating another team $j$, and there can be a circle among the match advantage relations.

In this paper, we do not assume such coherent ranking exists and solely rely on the *Borda score* based on preference probabilities. The Borda score $B(i)$ of an item $i$ is the probability that it is preferred when compared with another random item, namely $B(i) := \frac{1}{K-1} \sum_{j \neq i} p_{i,j}$. The item with the highest Borda score is called the *Borda winner*. The Borda winner is intuitively appealing and always well-defined for any set of preferential probabilities. The Borda score also does not require the problem instance to obey any consistency or transitivity, and it is considered one of the most general criteria.

To identify the Borda winner, estimations of the Borda scores are needed. Since estimating the Borda score for one item requires comparing it with every other items, the sample complexity is prohibitively high when there are numerous items. On the other hand, in many real-world applications, the agent has access to side information that can assist the evaluation of $p_{i,j}$. For instance, an e-commerce item carries its category as well as many other attributes, and the user might have a preference for a certain category (Wang et al., 2018). For a movie, the genre and the plot as well as the directors and actors can also be taken into consideration when making choices (Liu et al., 2017).

Based on the above motivation, we consider *Generalized Linear Dueling Bandits*. In each round, the agent selects two items from a finite set of items and receives a comparison result of the preferred item. The comparisons depend on known intrinsic contexts/features associated with each pair of items. The contexts can be obtained from upstream tasks, such as topic modeling (Zhu et al., 2012) or embedding (Vasile et al., 2016). Our goal is to adaptively select items and minimize the regret with respect to the optimal item (i.e., Borda winner). Our main contributions are summarized as follows:

- We show a hardness result regarding the Borda regret minimization for the (generalized) linear model. We prove a worst-case regret lower bound $\Omega(d^{2/3}T^{2/3})$ for our dueling bandit model, showing that even in the stochastic setting, minimizing the Borda regret is difficult. The construction and proof of the lower bound are new and might be of independent interest.
- We propose an explore-then-commit type algorithm under the stochastic setting, which can achieve a nearly matching upper bound $\widetilde{O}(d^{2/3}T^{2/3})$. When the number of items $K$ is small, the algorithm can also be configured to achieve a smaller regret $\widetilde{O}\big((d\log K)^{1/3}T^{2/3}\big)$.
- We propose an EXP3 type algorithm for linear dueling bandits under the adversarial setting, which can achieve a nearly matching upper bound $\widetilde{O}\big((d\log K)^{1/3}T^{2/3}\big)$.
- We conduct empirical studies to verify the correctness of our theoretical claims. Under both synthetic and real-world data settings, our algorithms can outperform all the baselines in terms of cumulative regret.

**Notation** In this paper, we use normal letters to denote scalars, lowercase bold letters to denote vectors, and uppercase bold letters to denote matrices. For a vector $\mathbf{x}$, $\|\mathbf{x}\|$ denotes its $\ell_2$-norm. The weighted $\ell_2$-norm associated with a positive-definite matrix $\mathbf{A}$ is defined as $\|\mathbf{x}\|_{\mathbf{A}} = \sqrt{\mathbf{x}^\top \mathbf{A} \mathbf{x}}$. The minimum eigenvalue of a matrix $\mathbf{A}$ is written as $\lambda_{\min}(\mathbf{A})$. We use $\mathbf{A} \succeq \mathbf{B}$ to denote that the matrix $\mathbf{A} - \mathbf{B}$ is positive semi-definite. We use standard asymptotic notations including $O(\cdot), \Omega(\cdot), \Theta(\cdot)$, and $\widetilde{O}(\cdot), \widetilde{\Omega}(\cdot), \widetilde{\Theta}(\cdot)$ will hide logarithmic factors. For a positive integer $N$, $[N] := \{1, 2, \dots, N\}$.

## 2 RELATED WORK

**Multi-armed and Contextual Bandits** Multi-armed bandit is a problem of identifying the best choice in a sequential decision-making system. It has been studied in numerous ways with a wide range of applications (Even-Dar et al., 2002; Lai et al., 1985; Kuleshov & Precup, 2014). Contextual linear bandit is a special type of bandit problem where the agent is provided with side information, i.e., contexts, and rewards are assumed to have a linear structure. Various algorithms (Rusmevichientong & Tsitsiklis, 2010; Filippi et al., 2010; Abbasi-Yadkori et al., 2011; Li et al., 2017; Jun et al., 2017) have been proposed to utilize this contextual information.

**Dueling Bandits and Its Performance Metrics** Dueling bandits is a variant of MAB with preferential feedback (Yue et al., 2012; Zoghi et al., 2014a; 2015). A comprehensive survey can be found at Bengs et al. (2021). As discussed previously, the probabilistic structure of a dueling bandits problem is governed by the preference probabilities, over which an optimal item needs to be defined. Optimality under the *Borda score* criteria has been adopted by several previous works (Jamieson et al., 2015; Falahatgar et al., 2017a; Heckel et al., 2018; Saha et al., 2021a). The most relevant work to ours is Saha et al. (2021a), where they studied the problem of regret minimization for adversarial dueling bandits and proved a $T$-round Borda regret upper bound $\widetilde{O}(K^{1/3}T^{2/3})$. They also provide an $\Omega(K^{1/3}T^{2/3})$ lower bound for stationary dueling bandits using Borda regret.

Apart from the Borda score, *Copeland score* is also a widely used criteria (Urvoy et al., 2013; Zoghi et al., 2015; 2014b; Wu & Liu, 2016; Komiyama et al., 2016). It is defined as $C(i) := \frac{1}{K-1}\sum_{j\neq i}\mathbb{1}\{p_{i,j} > 1/2\}$. A Copeland winner is the item that beats the most number of other items. It can be viewed as a "thresholded" version of Borda winner. In addition to Borda and Copeland winners, optimality notions such as a von Neumann winner were also studied in Ramamohan et al. (2016); Dudík et al. (2015); Balsubramani et al. (2016).

Another line of work focuses on identifying the optimal item or the total ranking, assuming the preference probabilities are consistent. Common consistency conditions include Strong Stochastic Transitivity (Yue et al., 2012; Falahatgar et al., 2017a;b), Weak Stochastic Transitivity (Falahatgar et al., 2018; Ren et al., 2019; Wu et al., 2022; Lou et al., 2022), Relaxed Stochastic Transitivity (Yue & Joachims, 2011) and Stochastic Triangle Inequality. Sometimes the aforementioned transitivity can also be implied by some structured models like the Bradley–Terry model. We emphasize that these consistency conditions are not assumed or implicitly implied in our setting.

**Contextual Dueling Bandits** In Dudík et al. (2015), contextual information is incorporated in the dueling bandits framework. Later, Saha (2021) studied a structured contextual dueling bandits setting where each item $i$ has its own contextual vector $\mathbf{x}_i$ (sometimes called Linear Stochastic Transitivity). Each item then has an intrinsic score $v_i$ equal to the linear product of an unknown parameter vector $\boldsymbol{\theta}^*$ and its contextual vector $\mathbf{x}_i$. The preference probability between two items $i$ and $j$ is assumed to be $\mu(v_i - v_j)$ where $\mu(\cdot)$ is the logistic function. These intrinsic scores of items naturally define a ranking over items. The regret is also computed as the gap between the scores of pulled items and the best item. While in this paper, we assume that the contextual vectors are associated with item pairs and define regret on the Borda score. In Section A.1, we provide a more detailed discussion showing that the setting considered in Saha (2021) can be viewed as a special case of our model.

## 3 BACKGROUNDS AND PRELIMINARIES

### 3.1 PROBLEM SETTING

We first consider the stochastic preferential feedback model with $K$ items in the fixed time horizon setting. We denote the item set by $[K]$ and let $T$ be the total number of rounds. in each round $t$, the agent can pick any pair of items $(i_t, j_t)$ to compare and receive stochastic feedback about whether item $i_t$ is preferred over item $j_t$, (denoted by $i_t \succ j_t$). We denote the probability of seeing the event $i \succ j$ as $p_{i,j} \in [0, 1]$. Naturally, we assume $p_{i,j} + p_{j,i} = 1$, and $p_{i,i} = 1/2$.

In this paper, we are concerned with the generalized linear model (GLM), where there is assumed to exist an *unknown* parameter $\boldsymbol{\theta}^* \in \mathbb{R}^d$, and each pair of items $(i, j)$ has its own *known* contextual/feature vector $\boldsymbol{\phi}_{i,j} \in \mathbb{R}^d$ with $\|\boldsymbol{\phi}_{i,j}\| \leq 1$. There is also a fixed known link function (sometimes called comparison function) $\mu(\cdot)$ that is monotonically increasing and satisfies $\mu(x) + \mu(-x) = 1$, e.g. a linear function or the logistic function $\mu(x) = 1/(1 + e^{-x})$. The preference probability is defined as $p_{i,j} = \mu(\boldsymbol{\phi}_{i,j}^\top \boldsymbol{\theta}^*)$. In each round, denote $r_t = \mathbb{1}\{i_t \succ j_t\}$, then we have

$$\mathbb{E}[r_t | i_t, j_t] = p_{i_t, j_t} = \mu(\boldsymbol{\phi}_{i_t, j_t}^\top \boldsymbol{\theta}^*).$$

Then our model can also be written as

$$r_t = \mu(\boldsymbol{\phi}_{i_t, j_t}^\top \boldsymbol{\theta}^*) + \epsilon_t,$$

where the noises $\{\epsilon_t\}_{t \in [T]}$ are zero-mean, 1-sub-Gaussian and assumed independent from each other. Note that, given the constraint $p_{i,j} + p_{j,i} = 1$, it is implied that $\phi_{i,j} = -\phi_{j,i}$ for any $i \in [K], j \in [K]$.

The agent's goal is to maximize the cumulative Borda score. The (slightly modified [1]) Borda score of item $i$ is defined as $B(i) = \frac{1}{K} \sum_{j=1}^{K} p_{i,j}$, and the Borda winner is defined as $i^* = \text{argmax}_{i \in [K]} B(i)$. The problem of merely identifying the Borda winner was deemed trivial (Zoghi et al., 2014a; Busa-Fekete et al., 2018) because for a fixed item $i$, uniformly random sampling $j$ and receiving feedback $r_{i,j} = \text{Bernoulli}(p_{i,j})$ yield a Bernoulli random variable with its expectation being the Borda score $B(i)$. This so-called *Borda reduction* trick makes identifying the Borda winner as easy as the best-arm identification for $K$-armed bandits. Moreover, if the regret is defined as $\text{Regret}(T) = \sum_{t=1}^{T} (B(i^*) - B(i_t))$, then any optimal algorithms for multi-arm bandits can achieve $\widetilde{O}(\sqrt{T})$ regret.

However, the above definition of regret does not respect the fact that a pair of items is selected in each round. When the agent chooses two items to compare, it is natural to define the regret so that both items contribute equally. A commonly used regret, e.g., in Saha et al. (2021a), has the following form:

$$\text{Regret}(T) = \sum_{t=1}^{T} \big(2B(i^*) - B(i_t) - B(j_t)\big), \tag{3.1}$$

where the regret is defined as the sum of the sub-optimality of both selected arms. Sub-optimality is measured by the gap between the Borda scores of the compared items and the Borda winner. This form of regret deems any classical multi-arm bandit algorithm with Borda reduction vacuous because taking $j_t$ into consideration will invoke $\Theta(T)$ regret.

**Adversarial Setting** Saha et al. (2021b) considered an adversarial setting for the multi-armed case, where in each round $t$, the comparison follows a potentially different probability model, denoted by $\{p_{i,j}^t\}_{i,j \in [K]}$. In this paper, we consider its contextual counterpart. Formally, we assume there is an underlying parameter $\boldsymbol{\theta}_t^*$, and in round $t$, the preference probability is defined as $p_{i,j}^t = \mu(\phi_{i,j}^\top \boldsymbol{\theta}_t^*)$.

The Borda score of item $i \in [K]$ in round $t$ is defined as $B_t(i) = \frac{1}{K} \sum_{j=1}^{K} p_{i,j}^t$, and the Borda winner in round $T$ is defined as $i^* = \text{argmax}_{i \in [K]} \sum_{t=1}^{T} B_t(i)$. The $T$-round regret is thus defined as $\text{Regret}(T) = \sum_{t=1}^{T} \big(2B_t(i^*) - B_t(i_t) - B_t(j_t)\big)$.

### 3.2 ASSUMPTIONS

In this section, we present the assumptions required for establishing theoretical guarantees. Due to the fact that the analysis technique is largely extracted from Li et al. (2017), we follow them to make assumptions to enable regret minimization for generalized linear dueling bandits.

We make a regularity assumption about the distribution of the contextual vectors:

**Assumption 3.1.** There exists a constant $\lambda_0 > 0$ such that $\lambda_{\min}\big(\frac{1}{K^2} \sum_{i=1}^{K} \sum_{j=1}^{K} \phi_{i,j} \phi_{i,j}^\top\big) \geq \lambda_0$.

This assumption is only utilized to initialize the design matrix $\mathbf{V}_\tau = \sum_{t=1}^{\tau} \phi_{i_t,j_t} \phi_{i_t,j_t}^\top$ so that the minimum eigenvalue is large enough. We follow Li et al. (2017) to deem $\lambda_0$ as a constant.

We also need the following assumption regarding the link function $\mu(\cdot)$:

**Assumption 3.2.** Let $\dot{\mu}$ be the first-order derivative of $\mu$. We have $\kappa := \inf_{\|\mathbf{x}\| \leq 1, \|\boldsymbol{\theta} - \boldsymbol{\theta}^*\| \leq 1} \dot{\mu}(\mathbf{x}^\top \boldsymbol{\theta}) > 0$.

Assuming $\kappa > 0$ is necessary to ensure the maximum log-likelihood estimator can converge to the true parameter $\boldsymbol{\theta}^*$ (Li et al., 2017, Section 3). This type of assumption is commonly made in previous works for generalized linear models (Filippi et al., 2010; Li et al., 2017; Faury et al., 2020).

---

[1]Previous works define Borda score as $B_i' = \frac{1}{K-1} \sum_{j \neq i} p_{i,j}$, excluding the diagonal term $p_{i,i} = 1/2$. Our definition is equivalent since the difference between two items satisfies $B(i) - B_j = \frac{K-1}{K}(B_i' - B_j')$. Therefore, the regret will be in the same order for both definitions.

Another common assumption is regarding the continuity and smoothness of the link function.

**Assumption 3.3.** $\mu$ is twice differentiable. Its first and second-order derivatives are upper-bounded by constants $L_\mu$ and $M_\mu$ respectively.

This is a very mild assumption. For example, it is easy to verify that the logistic link function satisfies Assumption 3.3 with $L_\mu = M_\mu = 1/4$.

## 4 THE HARDNESS RESULT

Figure 1: Illustration of the hard-to-learn preference probability matrix $\{p_{i,j}^{\boldsymbol{\theta}}\}_{i\in[K],j\in[K]}$. There are $K = 2^{d+1}$ items in total. The first $2^d$ items are "good" items with higher Borda scores, and the last $2^d$ items are "bad" items. The upper right block $\{p_{i,j}\}_{i<2^d,j\geq2^d}$ is defined as shown in the blue bubble. The lower left block satisfies $p_{i,j} = 1 - p_{j,i}$. For any $\boldsymbol{\theta}$, there exist one and only best item $i$ such that $\mathbf{bit}(i) = \mathbf{sign}(\boldsymbol{\theta})$.

This section presents Theorem 4.1, a worst-case regret lower bound for the stochastic linear dueling bandits. The proof of Theorem 4.1 relies on a class of hard instances, as shown in Figure 1. We show that any algorithm will incur a certain amount of regret when applied to this hard instance class. The constructed hard instances follow a stochastic linear model, which is a sub-class of the generalized linear model. Saha et al. (2021b) first proposed a similar construction for finite many arms with no contexts. Their design contains $K - 1$ identical arms against one best arm. This design will not work in our setting as it leads to lower bounds sub-optimal in dimension $d$. The new construction of our lower bound is based on the hardness of identifying the best arm in the $d$-dimensional linear bandit model and the proof of the lower bound takes a rather different route.

For any $d > 0$, we construct the class of hard instances as follows. An instance is specified by a vector $\boldsymbol{\theta} \in \{-\Delta, +\Delta\}^d$. The instance contains $2^{d+1}$ items (indexed from 0 to $2^{d+1} - 1$). The preference probability for an instance is defined by $p_{i,j}^{\boldsymbol{\theta}}$ as:

$$p_{i,j}^{\boldsymbol{\theta}} = \begin{cases} \frac{1}{2}, & \text{if } i < 2^d, j < 2^d \text{ or if } i \geq 2^d, j \geq 2^d \\ \frac{3}{4}, & \text{if } i < 2^d, j \geq 2^d \\ \frac{1}{4}, & \text{if } i \geq 2^d, j < 2^d \end{cases} + \langle \boldsymbol{\phi}_{i,j}, \boldsymbol{\theta} \rangle,$$

and the $d$-dimensional feature vectors $\boldsymbol{\phi}_{i,j}$ are given by

$$\boldsymbol{\phi}_{i,j} = \begin{cases} \mathbf{0}, & \text{if } i < 2^d, j < 2^d \text{ or if } i \geq 2^d, j \geq 2^d \\ \mathbf{bit}(i), & \text{if } i < 2^d, j \geq 2^d \\ -\mathbf{bit}(j), & \text{if } i \geq 2^d, j < 2^d, \end{cases}$$

where $\mathbf{bit}(\cdot)$ is the (shifted) bit representation of non-negative integers, i.e., suppose $x$ has the binary representation $x = b_0 \times 2^0 + b_1 \times 2^1 + \cdots + b_{d-1} \times 2^{d-1}$, then

$$\mathbf{bit}(x) = (2b_0 - 1, 2b_1 - 1, \ldots, 2b_{d-1} - 1) = 2\boldsymbol{b} - 1.$$

Note that $\mathbf{bit}(\cdot) \in \{-1, +1\}^d$, and that $\boldsymbol{\phi}_{i,j} = -\boldsymbol{\phi}_{j,i}$ is satisfied. The definition of $p_{i,j}^{\boldsymbol{\theta}}$ can be slightly tweaked to fit exactly the model described in Section 3 (see Remark B.1 in Appendix).

Some calculation shows that the Borda scores of the $2^{d+1}$ items are:

$$B^{\boldsymbol{\theta}}(i) = \begin{cases} \frac{5}{8} + \frac{1}{2}\langle \mathbf{bit}(i), \boldsymbol{\theta} \rangle, & \text{if } i < 2^d, \\ \frac{3}{8}, & \text{if } i \geq 2^d. \end{cases}$$

Intuitively, the former half of items (those indexed from 0 to $2^d - 1$) are "good" items (one among them is optimal, others are nearly optimal), while the latter half of items are "bad" items. Under such hard instances, every time one of the two pulled items is a "bad" item, then a one-step regret $B^{\boldsymbol{\theta}}(i^*) - B^{\boldsymbol{\theta}}(i) \geq 1/4$ is incurred. To minimize regret, we should thus try to avoid pulling "bad" items. However, in order to identify the best item among all "good" items, comparisons between "good" and "bad" items are necessary. The reason is simply that comparisons between "good" items give no information about the Borda scores as the comparison probabilities are $p_{i,j}^{\boldsymbol{\theta}} = \frac{1}{2}$ for all $i, j < 2^d$. Hence, any algorithm that can decently distinguish among the "good" items has to pull "bad" ones for a fair amount of times, and large regret is thus incurred. A similar observation is also made by Saha et al. (2021a).

This specific construction emphasizes the intrinsic hardness of Borda regret minimization: to differentiate the best item from its close competitors, the algorithm must query the bad items to gain information.

Formally, this class of hard instances leads to the following regret lower bound for both stochastic and adversarial settings:

**Theorem 4.1.** For any algorithm $\mathcal{A}$, there exists a hard instance $\{p_{i,j}^{\boldsymbol{\theta}}\}$ with $T > 4d^2$, such that $\mathcal{A}$ will incur expected regret at least $\Omega(d^{2/3}T^{2/3})$.

The construction of this hard instance for linear dueling bandits is inspired by the worst-case lower bound for the stochastic linear bandit (Dani et al., 2008), which has the order $\Omega(d\sqrt{T})$, while ours is $\Omega(d^{2/3}T^{2/3})$. The difference is that for the linear or multi-armed stochastic bandit, eliminating bad arms can make further exploration less expensive. But in our case, any amount of exploration will not reduce the cost of further exploration. This essentially means that exploration and exploitation must be separate, which is also supported by the fact that a simple explore-then-commit algorithm shown in Section 5 can be nearly optimal.

To prove the lower bound, we first apply a new reduction step to restrict the choice of $i_t$. Then we bound from below the regret by the expected number of sub-optimal arm pulls. The proof in Saha et al. (2021b) is directly based on hypothesis testing: either identifying the best arm with gap $\epsilon$ within $T$ rounds (if $T > \frac{K}{1440\epsilon^3}$) or incurring $\epsilon T$ regret (if $T \leq \frac{K}{1440\epsilon^3}$). In contrast, our proof technique bounds from below the regret by the expected number of sub-optimal arm pulls and does not divide the problem instances into two cases (i.e. whether $T \leq \frac{K}{1440\epsilon^3}$).

## 5 STOCHASTIC CONTEXTUAL DUELING BANDIT

### 5.1 ALGORITHM DESCRIPTION

We propose an algorithm named Borda Explore-Then-Commit for Generalized Linear Models (BETC-GLM), presented in Algorithm 1. Our algorithm is inspired by the algorithm for generalized linear models proposed by Li et al. (2017).

At the high level, Algorithm 1 can be divided into two phases: the exploration phase (Line 2-11) and the exploitation phase (Line 12-14). The exploration phase ensures that the MLE estimator $\widehat{\boldsymbol{\theta}}$ is accurate enough so that the estimated Borda score is within $\widetilde{O}(\epsilon)$-range of the true Borda score (ignoring other quantities). Then the exploitation phase simply chooses the empirical Borda winner to incur small regret.

During the exploration phase, the algorithm first performs "pure exploration" (Line 2-5), which can be seen as an initialization step for the algorithm. The purpose of this step is to ensure the design matrix $\mathbf{V}_{\tau+N} = \sum_{t=1}^{\tau+N} \boldsymbol{\phi}_{i_t,j_t} \boldsymbol{\phi}_{i_t,j_t}^{\top}$ is positive definite.

After that, the algorithm will perform the "designed exploration". Line 6 will find the G-optimal design, which minimizes the objective function $g(\pi) = \max_{i,j} \|\boldsymbol{\phi}_{i,j}\|_{\mathbf{V}(\pi)^{-1}}^2$, where $\mathbf{V}(\pi) := \sum_{i,j} \pi(i,j)\boldsymbol{\phi}_{i,j}\boldsymbol{\phi}_{i,j}^{\top}$. The G-optimal design $\pi^*(\cdot)$ satisfies $\|\boldsymbol{\phi}_{i,j}\|_{\mathbf{V}(\pi^*)^{-1}}^2 \leq d$, and can be efficiently approximated by the Frank-Wolfe algorithm (See Remark 5.4 for a detailed discussion). Then the algorithm will follow $\pi(\cdot)$ found at Line 6 to determine how many samples (Line 7) are needed.

---

**Algorithm 1** BETC-GLM

1: **Input:** time horizon $T$, number of items $K$, feature dimension $d$, feature vectors $\phi_{i,j}$ for $i \in [K]$, $j \in [K]$, exploration rounds $\tau$, error tolerance $\epsilon$, failure probability $\delta$.
2: **for** $t = 1, 2, \ldots, \tau$ **do**
3:     sample $i_t \sim \text{Uniform}([K])$, $j_t \sim \text{Uniform}([K])$
4:     query pair $(i_t, j_t)$ and receive feedback $r_t$
5: **end for**
6: Find the G-optimal design $\pi(i, j)$ based on $\phi_{i,j}$ for $i \in [K], j \in [K]$
7: Let $N(i, j) = \left\lceil \frac{d\pi(i,j)}{\epsilon^2} \right\rceil$ for any $(i, j) \in \text{supp}(\pi)$, denote $N = \sum_{i=1}^{K} \sum_{j=1}^{K} N(i, j)$
8: **for** $i \in [K]$, $j \in [K]$, $s \in [N(i, j)]$ **do**
9:     set $t \leftarrow t + 1$, set $(i_t, j_t) = (i, j)$
10:     query pair $(i_t, j_t)$ and receive feedback $r_t$
11: **end for**
12: Calculate the empirical MLE estimator $\widehat{\boldsymbol{\theta}}_{\tau+N}$ based on all $\tau + N$ samples via (5.1)
13: Estimate the Borda score for each item:

$$\widehat{B}(i) = \frac{1}{K} \sum_{j=1}^{K} \mu(\phi_{i,j}^{\top} \widehat{\boldsymbol{\theta}}_{\tau+N}), \qquad \widehat{i} = \underset{i \in [K]}{\text{argmax}} \, \widehat{B}(i)$$

14: Keep querying $(\widehat{i}, \widehat{i})$ for the rest of the time.

---

At Line 8-11, there are in total $N = \sum_{i=1}^{K} \sum_{j=1}^{K} N(i, j)$ samples queried, and the algorithm shall index them by $t = \tau + 1, \tau + 2, \ldots, \tau + N$.

At Line 12, the algorithm collects all the $\tau + N$ samples and performs the maximum likelihood estimation (MLE). For the generalized linear model, the MLE estimator $\widehat{\boldsymbol{\theta}}_{\tau+N}$ satisfies:

$$\sum_{t=1}^{\tau+N} \mu(\phi_{i_t,j_t}^{\top} \widehat{\boldsymbol{\theta}}_{\tau+N}) \phi_{i_t,j_t} = \sum_{t=1}^{\tau+N} r_t \phi_{i_t,j_t}, \tag{5.1}$$

or equivalently, it can be determined by solving a strongly concave optimization problem:

$$\widehat{\boldsymbol{\theta}}_{\tau+N} \in \underset{\boldsymbol{\theta}}{\text{argmax}} \sum_{t=1}^{\tau+N} \left( r_t \phi_{i_t,j_t}^{\top} \boldsymbol{\theta} - m(\phi_{i_t,j_t}^{\top} \boldsymbol{\theta}) \right),$$

where $\dot{m}(\cdot) = \mu(\cdot)$. For the logistic link function, $m(x) = \log(1 + e^x)$. As a special case of our generalized linear model, the linear model has a closed-form solution for (5.1). For example, if $\mu(x) = \frac{1}{2} + x$, i.e. $p_{i,j} = \frac{1}{2} + \phi_{i,j}^{\top} \boldsymbol{\theta}^*$, then (5.1) becomes:

$$\widehat{\boldsymbol{\theta}}_{\tau+N} = \mathbf{V}_{\tau+N}^{-1} \sum_{t=1}^{\tau+N} (r_t - 1/2) \phi_{i_t,j_t},$$

where $\mathbf{V}_{\tau+N} = \sum_{t=1}^{\tau+N} \phi_{i_t,j_t} \phi_{i_t,j_t}^{\top}$.

After the MLE estimator is obtained, Line 13 will calculate the estimated Borda score $\widehat{B}(i)$ for each item based on $\widehat{\boldsymbol{\theta}}_{\tau+N}$, and pick the empirically best one.

## 5.2 A MATCHING REGRET UPPER BOUND

Algorithm 1 can be configured to tightly match the worst-case lower bound. The configuration and performance are described as follows:

**Theorem 5.1.** Suppose Assumption 3.1-3.3 hold and $T = \Omega(d^2)$. For any $\delta > 0$, if we set $\tau = C_4 \lambda_0^{-2} (d + \log(1/\delta))$ ($C_4$ is a universal constant) and $\epsilon = d^{1/6} T^{-1/3}$, then with probability at least $1 - 2\delta$, Algorithm 1 will incur regret bounded by:

$$O\left( \kappa^{-1} d^{2/3} T^{2/3} \sqrt{\log(T/d\delta)} \right).$$

By setting $\delta = T^{-1}$, the expected regret is bounded as $\widetilde{O}(\kappa^{-1} d^{2/3} T^{2/3})$.

For linear bandit models, such as the hard-to-learn instances in Section 4, $\kappa$ is a universal constant. Therefore, Theorem 5.1 tightly matches the lower bound in Theorem 4.1, up to logarithmic factors. The detailed proof can be found in Appendix C.1.

**Remark 5.2** (Regret for Fewer Arms). In typical scenarios, the number of items $K$ is not exponentially large in the dimension $d$. In this case, we can choose a different parameter set of $\tau$ and $\epsilon$ such that Algorithm 1 can achieve a smaller regret bound $\widetilde{O}\big(\kappa^{-1}(d\log K)^{1/3}T^{2/3}\big)$ with smaller dependence on the dimension $d$. See Theorem A.1 in Appendix A.2.

**Remark 5.3** (Regret for Infinitely Many Arms). In most practical scenarios of dueling bandits, it is adequate to consider a finite number $K$ of items (e.g., ranking items). Nonetheless, BETC-GLM can be easily adapted to accommodate infinitely many arms in terms of regret. We can construct a covering over all $\phi_{i,j}$ and perform optimal design and exploration on the covering set. The resulting regret will be the same as our upper bound, i.e., $\widetilde{O}(d^{2/3}T^{2/3})$ up to some error caused by the epsilon net argument.

**Remark 5.4** (Approximate G-optimal Design). Algorithm 1 assumes an exact G-optimal design $\pi$ is obtained. In the experiments, we use the Frank-Wolfe algorithm to solve the constraint optimization problem (See Algorithm 5, Appendix G.3). To find a policy $\pi$ such that $g(\pi) \leq (1+\varepsilon)g(\pi^*)$, roughly $O(d/\varepsilon)$ optimization steps are needed. Such a near-optimal design will introduce a factor of $(1+\varepsilon)^{1/3}$ into the upper bounds.

## 6 Adversarial Contextual Dueling Bandit

This section addresses Borda regret minimization under the adversarial setting. As we introduced in Section 3.1, the unknown parameter $\theta_t$ can vary for each round $t$, while the contextual vectors $\phi_{i,j}$ are fixed.

Our proposed algorithm, BEXP3, is designed for the contextual linear model. Formally, in round $t$ and given pair $(i,j)$, we have $p_{i,j}^t = \frac{1}{2} + \langle \phi_{i,j}, \theta_t^* \rangle$.

### 6.1 Algorithm Description

---
**Algorithm 2** BEXP3
---

1: **Input:** time horizon $T$, number of items $K$, feature dimension $d$, feature vectors $\phi_{i,j}$ for $i \in [K]$, $j \in [K]$, learning rate $\eta$, exploration parameter $\gamma$.
2: **Initialize:** $q_1(i) = \frac{1}{K}$.
3: **for** $t = 1, \ldots, T$ **do**
4:      Sample items $i_t \sim q_t$, $j_t \sim q_t$.
5:      Query pair $(i_t, j_t)$ and receive feedback $r_t$
6:      Calculate $Q_t = \sum_{i \in [K]} \sum_{j \in [K]} q_t(i) q_t(j) \phi_{i,j} \phi_{i,j}^\top$, $\widehat{\theta}_t = Q_t^{-1} \phi_{i_t, j_t} r_t$.
7:      Calculate the (shifted) Borda score estimates $\widehat{B}_t(i) = \langle \frac{1}{K} \sum_{j \in [K]} \phi_{i,j}, \widehat{\theta}_t \rangle$.
8:      Update for all $i \in [K]$, set

$$\widetilde{q}_{t+1}(i) = \frac{\exp(\eta \sum_{l=1}^t \widehat{B}_l(i))}{\sum_{j \in [K]} \exp(\eta \sum_{l=1}^t \widehat{B}_l(j))}; \qquad q_{t+1}(i) = (1-\gamma)\widetilde{q}_{t+1}(i) + \frac{\gamma}{K}.$$

9: **end for**

---

Algorithm 2 is adapted from the DEXP3 algorithm in Saha et al. (2021b), which deals with the adversarial multi-armed dueling bandit. Algorithm 2 maintains a distribution $q_t(\cdot)$ over $[K]$, initialized as uniform distribution (Line 2). In every round $t$, two items are chosen following $q_t$ independently. Then Line 6 calculates the one-sample unbiased estimate $\widehat{\theta}_t$ of the true underlying parameter $\theta_t^*$. Line 7 further calculates the unbiased estimate of the (shifted) Borda score. Note that the true Borda score in round $t$ satisfies $B_t(i) = \frac{1}{2} + \langle \frac{1}{K} \sum_{j \in [K]} \phi_{i,j}, \theta_t^* \rangle$. $\widehat{B}_t$ instead only estimates the second term of the Borda score. This is a choice to simplify the proof. The cumulative estimated score $\sum_{l=1}^t \widehat{B}_l(i)$ can be seen as the estimated cumulative reward of item $i$ in round $t$. In Line 8, $q_{t+1}$ is defined by the classic exponential weight update, along with a uniform exploration policy controlled by $\gamma$.

## 6.2 UPPER BOUNDS

Algorithm 2 can also be configured to tightly match the worst-case lower bound:

**Theorem 6.1.** Suppose Assumption 3.1 holds. If we set $\eta = (\log K)^{2/3} d^{-1/3} T^{-2/3}$ and $\gamma = \sqrt{\eta d / \lambda_0} = (\log K)^{1/3} d^{1/3} T^{-1/3} \lambda_0^{-1/2}$, then the expected regret is upper-bounded by

$$O\big((d \log K)^{1/3} T^{2/3}\big).$$

Note that the lower bound construction in Theorem 4.1 is for the linear model and has $K = O(2^d)$, thus exactly matching the upper bound. Meanwhile, it is viable to slightly modify Algorithm 2 to improve the regret to $\widetilde{O}(d^{2/3} T^{2/3})$ for very large $K$. The high-level idea is to use an $\epsilon$-cover argument. In the $d$-dimensional space, it suffices to choose $O((1/\epsilon)^d)$ representative vectors to cover all the $K$ average contextual vectors $\frac{1}{K} \sum_{j=1}^{K} \phi_{i,j}$. The detailed reasoning and algorithm design can be found in Appendix H.

## 7 EXPERIMENTS

This section compares the proposed algorithm BETC-GLM with existing ones that are capable of minimizing Borda regret. We use random responses (generated from fixed preferential matrices) to interact with all tested algorithms. Each algorithm is run for 50 times over a time horizon of $T = 10^6$. We report both the mean and the standard deviation of the cumulative Borda regret and supply some analysis. The following list summarizes all methods we study in this section, a more complete description of the methods and parameters is available in Appendix E: BETC-GLM(-MATCH): Algorithm 1 proposed in this paper with different parameters. UCB-BORDA: The UCB al-

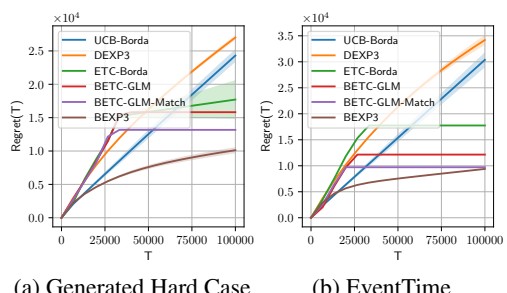

(a) Generated Hard Case      (b) EventTime

Figure 2: The regret of the proposed algorithms (BETC-GLM, BEXP3) and the baseline algorithms (UCB-BORDA, DEXP3, ETC-BORDA).

gorithm (Auer et al., 2002) using *Borda reduction*. DEXP3: Dueling-Exp3 developed by Saha et al. (2021a). ETC-BORDA: A simple explore-then-commit algorithm that does not take any contextual information into account. BEXP3: The proposed method for adversarial Borda bandits displayed in Algorithm 2.

**Generated Hard Case** We first test the algorithms on the hard instances constructed in Section 4. We generate $\theta^*$ randomly from $\{-\Delta, +\Delta\}^d$ with $\Delta = \frac{1}{4d}$ so that the comparison probabilities $p_{i,j}^{\theta^*} \in [0, 1]$ for all $i, j \in [K]$. We pick the dimension $d = 6$ and the number of arms is therefore $K = 2^{d+1} = 128$. Note the dual usage of $d$ in our construction and the model setup in Section 3.1. We refer readers to Remark B.1 in Appendix B for more details.

As depicted in Figure 2a, the proposed algorithms (BETC-GLM, BEXP3) outperform the baseline algorithms in terms of cumulative regret when reaching the end of time horizon $T$. For UCB-BORDA, since it is not tailored for the dueling regret definition, it suffers from a linear regret as its second arm is always sampled uniformly at random, leading to a constant regret per round. DEXP3 and ETC-BORDA are two algorithms designed for $K$-armed dueling bandits. Both are unable to utilize contextual information and thus demand more exploration. As expected, their regrets are higher than BETC-GLM or BEXP3. We do not fine-tune the hyper-parameters of both algorithms. In Appendix I, we show fine-tuning the error tolerance $\epsilon$ of BETC-GLM can further reduce the regret, and the two algorithms perform equally well.

**Real-world Dataset** To showcase the performance of the algorithms in a real-world setting, we use the EventTime dataset (Zhang et al., 2016). In this dataset, $K = 100$ historical events are compared in a pairwise fashion by crowd-sourced workers. We first calculate the empirical preference probabilities $\widetilde{p}_{i,j}$ from the collected responses, and construct a generalized linear model based on the

empirical preference probabilities. The algorithms are tested under this generalized linear model. Due to space limitations, more details are deferred to Appendix F.

As depicted in Figure 2b, the proposed algorithm BETC-GLM outperforms the baseline algorithms in terms of cumulative regret when reaching the end of time horizon $T$. The other proposed algorithm BEXP3 performs equally well even when misspecified (the algorithm is designed for the linear setting, while the comparison probability follows a logistic model).

## 8 CONCLUSION

In this paper, we introduced Borda regret into the generalized linear dueling bandits setting, along with an explore-then-commit type algorithm BETC-GLM and an EXP3 type algorithm BEXP3. The algorithms can achieve a nearly optimal regret upper bound, which we corroborate with a matching lower bound. The theoretical performance of the algorithms is verified empirically. It demonstrates superior performance compared to other baseline methods.

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

# A  ADDITIONAL RESULTS AND DISCUSSION

## A.1  EXISTING RESULTS FOR STRUCTURED CONTEXTS

A structural assumption made by some previous works (Saha, 2021) is that $\phi_{i,j} = \mathbf{x}_i - \mathbf{x}_j$, where $\mathbf{x}_i$ can be seen as some feature vectors tied to the item. In this work, we do not consider minimizing the Borda regret under the structural assumption.

The immediate reason is that, when $p_{i,j} = \mu(\mathbf{x}_i^\top \boldsymbol{\theta}^* - \mathbf{x}_j^\top \boldsymbol{\theta}^*)$, with $\mu(\cdot)$ being the logistic function, the probability model $p_{i,j}$ effectively becomes (a linear version of) the well-known Bradley-Terry model. Namely, each item is tied to a value $v_i = \mathbf{x}_i^\top \boldsymbol{\theta}^*$, and the comparison probability follows $p_{i,j} = \frac{e^{v_i}}{e^{v_i} + e^{v_j}}$. More importantly, this kind of model satisfies both the strong stochastic transitivity (SST) and the stochastic triangle inequality (STI), which are unlikely to satisfy in reality.

Furthermore, when stochastic transitivity holds, there is a true ranking among the items, determined by $\mathbf{x}_i^\top \boldsymbol{\theta}^*$. A true ranking renders concepts like the Borda winner or Copeland winner redundant because the rank-one item will always be the winner in every sense. When $\phi_{i,j} = \mathbf{x}_i - \mathbf{x}_j$, Saha (2021) proposed algorithms that can achieve nearly optimal regret $\widetilde{O}(d\sqrt{T})$, with regret being defined as

$$\text{Regret}(T) = \sum_{t=1}^{T} 2\langle \mathbf{x}_{i^*}, \boldsymbol{\theta}^* \rangle - \langle \mathbf{x}_{i_t}, \boldsymbol{\theta}^* \rangle - \langle \mathbf{x}_{j_t}, \boldsymbol{\theta}^* \rangle, \tag{A.1}$$

where $i^* = \operatorname{argmax}_i \langle \mathbf{x}_i, \boldsymbol{\theta}^* \rangle$, which also happens to be the Borda winner. Meanwhile, by Assumption 3.3,

$$B(i^*) - B(j) = \frac{1}{K} \sum_{k=1}^{K} \left[ \mu(\langle \mathbf{x}_{i^*} - \mathbf{x}_k, \boldsymbol{\theta}^* \rangle) - \mu(\langle \mathbf{x}_j - \mathbf{x}_k, \boldsymbol{\theta}^* \rangle) \right] \leq L_\mu \cdot \langle \mathbf{x}_{i^*} - \mathbf{x}_j, \boldsymbol{\theta}^* \rangle,$$

where $L_\mu$ is the upper bound on the derivative of $\mu(\cdot)$. For logistic function $L_\mu = 1/4$. The Borda regret (3.1) is thus at most a constant multiple of (A.1). This shows Borda regret minimization can be sufficiently solved by Saha (2021) when structured contexts are present. We consider the most general case where the only restriction is the implicit assumption that $\phi_{i,j} = -\phi_{j,i}$.

## A.2  REGRET BOUND FOR FEWER ARMS

In typical scenarios, the number of items $K$ is not exponentially large in the dimension $d$. If this is the case, then we can choose a different parameter set of $\tau$ and $\epsilon$ such that Algorithm 1 can achieve a regret bound depending on $\log K$, and reduce the dependence on $d$. The performance can be characterized by the following theorem:

**Theorem A.1.** For any $\delta > 0$, suppose the number of total rounds $T$ satisfies,

$$T \geq \frac{C_3}{\kappa^6 \lambda_0^{3/2}} \max \left\{ d^{5/2}, \frac{\log(K^2/\delta)}{\sqrt{d}} \right\}, \tag{A.2}$$

where $C_3$ is some large enough universal constant. Then if we set $\tau = (d \log(K/\delta))^{1/3} T^{2/3}$ and $\epsilon = d^{1/3} T^{-1/3} \log(3K^2/\delta)^{-1/6}$, Algorithm 1 will incur regret bounded by:

$$O\big(\kappa^{-1}(d \log(K/\delta))^{1/3} T^{2/3}\big).$$

By setting $\delta = T^{-1}$, the expected regret is bounded as $\widetilde{O}\big(\kappa^{-1}(d \log K)^{1/3} T^{2/3}\big)$.

The detailed proof can be found in Appendix C.2.

# B  OMITTED PROOF IN SECTION 4

The proof relies on a class of hard-to-learn instances. We first present the construction again for completeness.

For any $d > 0$, we construct a hard instance with $2^{d+1}$ items (indexed from 0 to $2^{d+1} - 1$). We construct the hard instance $p_{i,j}^{\boldsymbol{\theta}}$ for any $\boldsymbol{\theta} \in \{-\Delta, +\Delta\}^d$ as:

$$p_{i,j}^{\boldsymbol{\theta}} = \begin{cases} \frac{1}{2}, & \text{if } i < 2^d, j < 2^d \\ \frac{1}{2}, & \text{if } i \geq 2^d, j \geq 2^d \\ \frac{3}{4}, & \text{if } i < 2^d, j \geq 2^d \\ \frac{1}{4}, & \text{if } i \geq 2^d, j < 2^d \end{cases} + \langle \boldsymbol{\phi}_{i,j}, \boldsymbol{\theta} \rangle, \tag{B.1}$$

where the feature vectors $\boldsymbol{\phi}_{i,j}$ and the parameter $\boldsymbol{\theta}$ are of dimension $d$, and have the following forms:

$$\boldsymbol{\phi}_{i,j} = \begin{cases} \mathbf{0}, & \text{if } i < 2^d, j < 2^d \\ \mathbf{0}, & \text{if } i \geq 2^d, j \geq 2^d \\ \mathbf{bit}(i), & \text{if } i < 2^d, j \geq 2^d \\ -\mathbf{bit}(j), & \text{if } i \geq 2^d, j < 2^d, \end{cases}$$

where $\mathbf{bit}(\cdot)$ is the (shifted) bit representation of non-negative integers, i.e., suppose $x = b_0 \times 2^0 + b_1 \times 2^1 + \cdots + b_{d-1} \times 2^{d-1}$, then $\mathbf{bit}(x) = 2\boldsymbol{b} - 1$. Note that $\mathbf{bit}(\cdot) \in \{-1, +1\}^d$, and $\boldsymbol{\phi}_{i,j} = -\boldsymbol{\phi}_{j,i}$.

**Remark B.1** ($d + 1$-dimensional instance). The hard instance described above does not strictly satisfy the assumption that $p_{i,j}^{\boldsymbol{\theta}} = \langle \boldsymbol{\theta}, \boldsymbol{\phi}_{i,j} \rangle$, but can be easily fixed by appending an additional dimension to address the bias term defined in (B.1). More specifically, we can set $F(x) = \frac{1}{2} + x$ and $p_{i,j}^{\boldsymbol{\theta}} = F(\langle \widetilde{\boldsymbol{\phi}}_{i,j}, \widetilde{\boldsymbol{\theta}} \rangle)$, where $\widetilde{\boldsymbol{\theta}} \in \{-\Delta, +\Delta\}^d \times \{\frac{1}{4}\} \subset \mathbb{R}^{d+1}$ and $\widetilde{\boldsymbol{\phi}}_{i,j} = (\boldsymbol{\phi}_{i,j}, c_{i,j})$, with $c_{i,j} = \begin{cases} 0, & \text{if } i < 2^d, j < 2^d \\ 0, & \text{if } i \geq 2^d, j \geq 2^d \\ 1, & \text{if } i < 2^d, j \geq 2^d \\ -1, & \text{if } i \geq 2^d, j < 2^d. \end{cases}$ To ensure $\|\widetilde{\boldsymbol{\phi}}_{i,j}\|_2 \leq 1$, we can further set $\widetilde{\boldsymbol{\phi}}_{i,j} \leftarrow (d+1)^{-1/2} \widetilde{\boldsymbol{\phi}}_{i,j}$ and $\widetilde{\boldsymbol{\theta}} \leftarrow (d+1)^{1/2} \widetilde{\boldsymbol{\theta}}$.

We rewrite (B.1) as:

$$p_{i,j}^{\boldsymbol{\theta}} = \begin{cases} \frac{1}{2}, & \text{if } i < 2^d, j < 2^d \\ \frac{1}{2}, & \text{if } i \geq 2^d, j \geq 2^d \\ \frac{3}{4}, & \text{if } i < 2^d, j \geq 2^d \\ \frac{1}{4}, & \text{if } i \geq 2^d, j < 2^d \end{cases} + \begin{cases} 0, & \text{if } i < 2^d, j < 2^d \\ 0, & \text{if } i \geq 2^d, j \geq 2^d \\ \langle \mathbf{bit}(i), \boldsymbol{\theta} \rangle, & \text{if } i < 2^d, j \geq 2^d \\ -\langle \mathbf{bit}(j), \boldsymbol{\theta} \rangle, & \text{if } i \geq 2^d, j < 2^d, \end{cases} \tag{B.2}$$

and the Borda scores are:

$$B^{\boldsymbol{\theta}}(i) = \begin{cases} \frac{5}{8} + \frac{1}{2} \langle \mathbf{bit}(i), \boldsymbol{\theta} \rangle, & \text{if } i < 2^d, \\ \frac{3}{8}, & \text{if } i \geq 2^d. \end{cases}$$

Intuitively, the former half arms indexed from 0 to $2^d - 1$ are "good" arms (one among them is optimal), while the latter half arms are "bad" arms. It is clear that choosing a "bad" arm $i$ will incur regret $B(i^*) - B(i) \geq 1/4$.

Now we are ready to present the proof.

*Proof of Theorem 4.1.* First, we present the following lemma:

**Lemma B.2.** Under the hard instance we constructed above, for any algorithm $\mathcal{A}$ that ever makes queries $i_t \geq 2^d$, there exists another algorithm $\mathcal{A}'$ that only makes queries $i_t < 2^d$ for every $t > 0$ and always achieves no larger regret than $\mathcal{A}$.

*Proof of Lemma B.2.* The proof is done by reduction. For any algorithm $\mathcal{A}$, we wrap $\mathcal{A}$ with such a agent $\mathcal{A}'$:

1. If $\mathcal{A}$ queries $(i_t, j_t)$ with $i_t < 2^d$, the agent $\mathcal{A}'$ will pass the same query $(i_t, j_t)$ to the environment and send the feedback $r_t$ to $\mathcal{A}$;

2. If $\mathcal{A}$ queries $(i_t, j_t)$ with $i_t \geq 2^d, j_t < 2^d$, the agent $\mathcal{A}'$ will pass the query $(j_t, i_t)$ to the environment and send the feedback $1 - r_t$ to $\mathcal{A}$;

3. If $\mathcal{A}$ queries $(i_t, j_t)$ with $i_t \geq 2^d, j_t \geq 2^d$, the agent $\mathcal{A}'$ will uniform-randomly choose $i'_t$ from 0 to $2^d - 1$, pass the query $(i'_t, i'_t)$ to the environment and send the feedback $r_t$ to $\mathcal{A}$.

For each of the cases defined above, the probabilistic model of bandit feedback for $\mathcal{A}$ is the same as if $\mathcal{A}$ is directly interacting with the original environment. For Case 1, the claim is trivial. For Case 2, the claim holds because of the symmetry of our model, that is $p_{i,j}^{\boldsymbol{\theta}} = 1 - p_{j,i}^{\boldsymbol{\theta}}$. For Case 3, both will return $r_t$ following Bernoulli(1/2). Therefore, the expected regret of $\mathcal{A}$ in this environment wrapped by $\mathcal{A}'$ is equal to the regret of $\mathcal{A}$ in the original environment.

Meanwhile, we will show $\mathcal{A}'$ will incur no larger regret than $\mathcal{A}$. For the first two cases, $\mathcal{A}'$ will incur the same one-step regret as $\mathcal{A}$. For the third case, we know that $B^{\boldsymbol{\theta}}(i_t) = B^{\boldsymbol{\theta}}(j_t) = \frac{3}{8}$, while $\mathbb{E}[B^{\boldsymbol{\theta}}(i'_t)] = \frac{5}{8} + \frac{1}{2}\langle \mathbb{E}_{i'_t}[\mathbf{bit}(i'_t)], \boldsymbol{\theta} \rangle = \frac{5}{8} + \frac{1}{2}\langle \mathbf{0}, \boldsymbol{\theta} \rangle = \frac{5}{8}$, meaning that the one-step regret is smaller. $\qquad \square$

Lemma B.2 ensures it is safe to assume $i_t < 2^d$. For any $\boldsymbol{\theta}$ and $k \in [d]$, define

$$\mathbb{P}_{\boldsymbol{\theta}, k} := \mathbb{P}_{\boldsymbol{\theta}}\left( \sum_{t=1}^{T} \mathbb{1}\{\mathbf{bit}^{[k]}(i_t) \neq \text{sign}(\boldsymbol{\theta}^{[k]})\} \geq \frac{T}{2} \right),$$

where the superscript $^{[k]}$ over a vector denotes taking the $k$-th entry of the vector. Meanwhile, we define $\boldsymbol{\theta}^{\backslash k}$ to satisfy $(\boldsymbol{\theta}^{\backslash k})^{[k]} = -\boldsymbol{\theta}^{[k]}$ and be the same as $\boldsymbol{\theta}$ at all other entries. We have

$$\mathbb{P}_{\boldsymbol{\theta}^{\backslash k}, k} := \mathbb{P}_{\boldsymbol{\theta}^{\backslash k}}\left( \sum_{t=1}^{T} \mathbb{1}\left\{ \mathbf{bit}^{[k]}(i_t) \neq \text{sign}\big((\boldsymbol{\theta}^{\backslash k})^{[k]}\big) \right\} \geq \frac{T}{2} \right)$$

$$= \mathbb{P}_{\boldsymbol{\theta}^{\backslash k}}\left( \sum_{t=1}^{T} \mathbb{1}\{\mathbf{bit}^{[k]}(i_t) = \text{sign}(\boldsymbol{\theta}^{[k]})\} \geq \frac{T}{2} \right)$$

$$= \mathbb{P}_{\boldsymbol{\theta}^{\backslash k}}\left( \sum_{t=1}^{T} \mathbb{1}\{\mathbf{bit}^{[k]}(i_t) \neq \text{sign}(\boldsymbol{\theta}^{[k]})\} < \frac{T}{2} \right).$$

Denote $\mathbb{P}_{\boldsymbol{\theta}, \mathcal{A}}(i_1, j_1, r_1, i_2, j_2, r_2, \dots)$ as the canonical probability distribution of algorithm $\mathcal{A}$ under the model $\mathbb{P}_{\boldsymbol{\theta}}$. By the Bretagnolle–Huber inequality and the decomposition of the relative entropy, we have

$$\mathbb{P}_{\boldsymbol{\theta}, k} + \mathbb{P}_{\boldsymbol{\theta}^{\backslash k}, k} \geq \frac{1}{2} \exp\big( -\text{KL}(\mathbb{P}_{\boldsymbol{\theta}, \mathcal{A}} \| \mathbb{P}_{\boldsymbol{\theta}^{\backslash k}, \mathcal{A}}) \big)$$

$$\geq \frac{1}{2} \exp\left( -\mathbb{E}_{\boldsymbol{\theta}}\left[ \sum_{t=1}^{T} \text{KL}\left( p_{i,j}^{\boldsymbol{\theta}} \,\Big\|\, p_{i,j}^{\boldsymbol{\theta}^{\backslash k}} \right) \right] \right)$$

$$\geq \frac{1}{2} \exp\left( -\mathbb{E}_{\boldsymbol{\theta}}\left[ \sum_{t=1}^{T} 10\langle \boldsymbol{\phi}_{i_t, j_t}, \boldsymbol{\theta} - \boldsymbol{\theta}^{\backslash k} \rangle^2 \right] \right)$$

$$= \frac{1}{2} \exp\left( -\mathbb{E}_{\boldsymbol{\theta}}\left[ 40\Delta^2 \sum_{t=1}^{T} \mathbb{1}\{i_t < 2^d \wedge j_t \geq 2^d\} \right] \right),$$

where the first inequality comes from the Bretagnolle–Huber inequality; the second inequality is the decomposition of the relative entropy; the third inequality holds because the Bernoulli KL divergence $KL(p \| p + x)$ is 10-strongly convex in $x$ for any fixed $p \in [1/8, 7/8]$, and indeed $p_{i,j}^{\boldsymbol{\theta}} \in [1/8, 7/8]$ as long as $d\Delta \leq 1/8$; the last equation holds because $\boldsymbol{\phi}_{i_t, j_t}$ has non-zero entries only when $(i_t, j_t)$ belongs to that specific regions.

From now on, we denote $N(T) := \sum_{t=1}^{T} \mathbb{1}\{i_t < 2^d \wedge j_t \geq 2^d\}$. Further averaging over all $\boldsymbol{\theta} \in \{-\Delta, +\Delta\}^d$, we have

$$\frac{1}{2^d} \sum_{\boldsymbol{\theta} \in \{-\Delta, +\Delta\}^d} \mathbb{P}_{\boldsymbol{\theta}, k} \geq \frac{1}{4} \frac{1}{2^d} \sum_{\boldsymbol{\theta} \in \{-\Delta, +\Delta\}^d} \exp\big( -40\Delta^2 \mathbb{E}_{\boldsymbol{\theta}}[N(T)] \big)$$

$$\geq \frac{1}{4} \exp\left( -40\Delta^2 \frac{1}{2^d} \sum_{\boldsymbol{\theta} \in \{-\Delta, +\Delta\}^d} \mathbb{E}_{\boldsymbol{\theta}}[N(T)] \right),$$

where the first inequality is from averaging over all $\boldsymbol{\theta}$; the second inequality is from Jensen's inequality.

Utilizing the inequality above, we establish that

$$
\begin{aligned}
\frac{1}{2^d} \sum_{\boldsymbol{\theta} \in \{-\Delta, +\Delta\}^d} \mathrm{Regret}(T; \boldsymbol{\theta}, \mathcal{A}) &\geq \frac{1}{2^d} \sum_{\boldsymbol{\theta} \in \{-\Delta, +\Delta\}^d} \mathbb{E}_{\boldsymbol{\theta}} \left[ \sum_{t=1}^{T} B^{\boldsymbol{\theta}}(i^*) - B^{\boldsymbol{\theta}}(i_t) \right] \\
&= \frac{1}{2^d} \sum_{\boldsymbol{\theta} \in \{-\Delta, +\Delta\}^d} \mathbb{E}_{\boldsymbol{\theta}} \left[ \sum_{t=1}^{T} \langle \boldsymbol{\theta}, \mathbf{sign}(\boldsymbol{\theta}) - \mathbf{bit}(i_t) \rangle \right] \\
&= \frac{1}{2^d} \sum_{\boldsymbol{\theta} \in \{-\Delta, +\Delta\}^d} \mathbb{E}_{\boldsymbol{\theta}} \left[ \sum_{t=1}^{T} \sum_{k=1}^{d} 2\Delta \, \mathbb{1}\{\mathbf{bit}^{[k]}(i_t) \neq \mathrm{sign}(\boldsymbol{\theta}^{[k]})\} \right] \\
&= \frac{2\Delta}{2^d} \sum_{\boldsymbol{\theta} \in \{-\Delta, +\Delta\}^d} \sum_{k=1}^{d} \mathbb{E}_{\boldsymbol{\theta}} \left[ \sum_{t=1}^{T} \mathbb{1}\{\mathbf{bit}^{[k]}(i_t) \neq \mathrm{sign}(\boldsymbol{\theta}^{[k]})\} \right] \\
&\geq \frac{2\Delta}{2^d} \sum_{\boldsymbol{\theta} \in \{-\Delta, +\Delta\}^d} \sum_{k=1}^{d} \mathbb{P}_{\boldsymbol{\theta}, k} \cdot \frac{T}{2} \\
&\geq \frac{\Delta d T}{4} \exp \left( -40\Delta^2 \frac{1}{2^d} \sum_{\boldsymbol{\theta} \in \{-\Delta, +\Delta\}^d} \mathbb{E}_{\boldsymbol{\theta}}[N(T)] \right), \quad \text{(B.3)}
\end{aligned}
$$

where the first inequality comes from the Borda regret; the second inequality comes from the inequality $\mathbb{E}[X] \geq a\mathbb{P}(X \geq a)$ for any non-negative random variable; the last inequality is from rearranging terms and invoking the results above.

Meanwhile, we have (remember $N(T) := \sum_{t=1}^{T} \mathbb{1}\{i_t < 2^d \wedge j_t \geq 2^d\}$)

$$
\begin{aligned}
\frac{1}{2^d} \sum_{\boldsymbol{\theta} \in \{-\Delta, +\Delta\}^d} \mathrm{Regret}(T; \boldsymbol{\theta}, \mathcal{A}) &\geq \frac{1}{2^d} \sum_{\boldsymbol{\theta} \in \{-\Delta, +\Delta\}^d} \mathbb{E}_{\boldsymbol{\theta}} \left[ \frac{1}{4} \sum_{t=1}^{T} \mathbb{1}\{i_t < 2^d \wedge j_t \geq 2^d\} \right] \\
&= \frac{1}{4} \frac{1}{2^d} \sum_{\boldsymbol{\theta} \in \{-\Delta, +\Delta\}^d} \mathbb{E}_{\boldsymbol{\theta}}[N(T)], \quad \text{(B.4)}
\end{aligned}
$$

where the first inequality comes from that any items $i \geq 2^d$ will incur at least $1/4$ regret.

Combining (B.3) and (B.4) together and denoting that $X = \frac{1}{2^d} \sum_{\boldsymbol{\theta} \in \{-\Delta, +\Delta\}^d} \mathbb{E}_{\boldsymbol{\theta}}[N(T)]$, we have that for any algorithm $\mathcal{A}$, there exists some $\boldsymbol{\theta}$, such that (set $\Delta = \frac{d^{-1/3} T^{-1/3}}{\sqrt{40}}$)

$$
\begin{aligned}
\mathrm{Regret}(T; \boldsymbol{\theta}, \mathcal{A}) &\geq \max \left\{ \frac{\Delta d T}{4} \exp(-40\Delta^2 X), \frac{X}{4} \right\} \\
&= \max \left\{ \frac{d^{2/3} T^{2/3}}{4\sqrt{40}} \exp(-d^{-2/3} T^{-2/3} X), \frac{X}{4} \right\} \\
&\geq \frac{d^{2/3} T^{2/3}}{4\sqrt{40}} \max \left\{ \exp(-d^{-2/3} T^{-2/3} X), d^{-2/3} T^{-2/3} X \right\} \\
&\geq \frac{d^{2/3} T^{2/3}}{8\sqrt{40}},
\end{aligned}
$$

where the first inequality is the combination of (B.3) and (B.4); the second inequality is a rearrangement and loosely lower bounds the constant; the last is due to $\max\{e^{-y}, y\} > 1/2$ for any $y$. $\qquad \square$

## C  Omitted Proof in Section 5

We first introduce the lemma about the theoretical guarantee of G-optimal design: given an action set $\mathcal{X} \subseteq \mathbb{R}^d$ that is compact and $\mathrm{span}(\mathcal{X}) = \mathbb{R}^d$. A fixed design $\pi(\cdot) : \mathcal{X} \to [0,1]$ satisfies $\sum_{\mathbf{x} \in \mathcal{X}} \pi(\mathbf{x}) = 1$. Define $\mathbf{V}(\pi) := \sum_{\mathbf{x} \in \mathcal{X}} \pi(\mathbf{x}) \mathbf{x} \mathbf{x}^\top$ and $g(\pi) := \max_{\mathbf{x} \in \mathcal{X}} \|\mathbf{x}\|_{\mathbf{V}(\pi)^{-1}}^2$.

**Lemma C.1** (The Kiefer–Wolfowitz Theorem, Section 21.1, Lattimore & Szepesvári (2020)). There exists an optimal design $\pi^*(\cdot)$ such that $|\mathrm{supp}(\pi)| \leq d(d+1)/2$, and satisfies:

1. $g(\pi^*) = d$.

2. $\pi^*$ is the minimizer of $g(\cdot)$.

The following lemma is also useful to show that under mild conditions, the minimum eigenvalue of the design matrix can be lower-bounded:

**Lemma C.2** (Proposition 1, Li et al. 2017). Define $\mathbf{V}_\tau = \sum_{t=1}^\tau \phi_{i_t,j_t} \phi_{i_t,j_t}^\top$, where each $(i_t, j_t)$ is drawn i.i.d. from some distribution $\nu$. Suppose $\lambda_{\min}\big(\mathbb{E}_{(i,j)\sim\nu}[\phi_{i,j}^\top \phi_{i,j}]\big) \geq \lambda_0$, and

$$\tau \geq \left( \frac{C_1 \sqrt{d} + C_2 \sqrt{\log(1/\delta)}}{\lambda_0} \right)^2 + \frac{2B}{\lambda_0},$$

where $C_1$ and $C_2$ are some universal constants. Then with probability at least $1 - \delta$,

$$\lambda_{\min}(\mathbf{V}_\tau) \geq B.$$

### C.1  Proof of Theorem 5.1

The proof relies on the following lemma to establish an upper bound on $|\langle \phi_{i,j}, \widehat{\boldsymbol{\theta}}_{\tau+N} - \boldsymbol{\theta}^* \rangle|$.

**Lemma C.3** (extracted from Lemma 3, Li et al. (2017)). Suppose $\lambda_{\min}(\mathbf{V}_{\tau+N}) \geq 1$. For any $\delta > 0$, with probability at least $1 - \delta$, we have

$$\|\widehat{\boldsymbol{\theta}}_{\tau+N} - \boldsymbol{\theta}^*\|_{\mathbf{V}_{\tau+N}} \leq \frac{1}{\kappa} \sqrt{\frac{d}{2} \log(1 + 2(\tau+N)/d) + \log(1/\delta)}.$$

*Proof of Theorem 5.1.* The proof can be divided into three steps: 1. invoke Lemma C.2 to show that the initial $\tau$ rounds for exploration will guarantee $\lambda_{\min}(\mathbf{V}_\tau) \geq 1$; 2. invoke Lemma C.1 to obtain an optimal design $\pi$ and utilize Cauchy-Schwartz inequality to show that $|\langle \widehat{\boldsymbol{\theta}}_{\tau+N} - \boldsymbol{\theta}, \phi_{i,j} \rangle| \leq 3\epsilon/\kappa$; 3. balance the not yet determined $\epsilon$ to obtain the regret upper bound.

Since we set $\tau$ such that

$$\tau = C_4 \lambda_0^{-2}(d + \log(1/\delta))$$
$$\geq \left( \frac{C_1 \sqrt{d} + C_2 \sqrt{\log(1/\delta)}}{\lambda_0} \right)^2 + \frac{2}{\lambda_0},$$

with a large enough universal constant $C_4$, by Lemma C.2 to obtain that with probability at least $1 - \delta$,

$$\lambda_{\min}(\mathbf{V}_\tau) \geq 1. \tag{C.1}$$

From now on, we assume (C.1) always holds.

Define $N := \sum_{i,j} N(i,j)$, $\mathbf{V}_{\tau+1:\tau+N} := \sum_{t=\tau+1}^{\tau+N} \phi_{i_t,j_t} \phi_{i_t,j_t}^\top$, $\mathbf{V}_{\tau+N} := \mathbf{V}_\tau + \mathbf{V}_{\tau+1:\tau+N}$. Given the optimal design $\pi(i,j)$, the algorithm queries the pair $(i,j) \in \mathrm{supp}(\pi)$ for exactly $N(i,j) = \lceil d\pi(i,j)/\epsilon^2 \rceil$ times. Therefore, the design matrix $\mathbf{V}_{\tau+N}$ satisfies

$$\mathbf{V}_{\tau+N} \succeq \mathbf{V}_{\tau+1:\tau+N}$$
$$= \sum_{i,j} N(i,j) \phi_{i,j} \phi_{i,j}^\top$$
$$\succeq \sum_{i,j} \frac{d\pi(i,j)}{\epsilon^2} \phi_{i,j} \phi_{i,j}^\top$$
$$= \frac{d}{\epsilon^2} \mathbf{V}(\pi),$$

where $\mathbf{V}(\pi) := \sum_{i,j} \pi(i,j)\phi_{i,j}\phi_{i,j}^\top$. The first inequality holds because $\mathbf{V}_\tau$ is positive semi-definite, and the second inequality holds due to the choice of $N(i,j)$.

When (C.1) holds, from Lemma C.3, we have with probability at least $1 - \delta$, that for each $\phi_{i,j}$,

$$
\begin{aligned}
|\langle \widehat{\boldsymbol{\theta}} - \boldsymbol{\theta}^*, \phi_{i,j} \rangle| &\leq \|\widehat{\boldsymbol{\theta}}_{\tau+N} - \boldsymbol{\theta}^*\|_{\mathbf{V}_{\tau+N}} \cdot \|\phi_{i,j}\|_{\mathbf{V}_{\tau+N}^{-1}} \\
&\leq \|\widehat{\boldsymbol{\theta}}_{\tau+N} - \boldsymbol{\theta}^*\|_{\mathbf{V}_{\tau+N}} \cdot \frac{\epsilon\|\phi_{i,j}\|_{\mathbf{V}(\pi)^{-1}}}{\sqrt{d}} \\
&\leq \|\widehat{\boldsymbol{\theta}}_{\tau+N} - \boldsymbol{\theta}^*\|_{\mathbf{V}_{\tau+N}} \cdot \epsilon \\
&\leq \frac{\epsilon}{\kappa} \cdot \sqrt{\frac{d}{2}\log(1 + 2(\tau+N)/d) + \log(1/\delta)}
\end{aligned}
\tag{C.2}
$$

where the first inequality is due to the Cauchy-Schwartz inequality; the second inequality holds because $\mathbf{V}_{\tau+N} \succeq \frac{d}{\epsilon^2}\mathbf{V}(\pi)$; the third inequality holds because $\pi$ is an optimal design and by Lemma C.1, $\|\phi_{i,j}\|_{\mathbf{V}(\pi)^{-1}}^2 \leq d$; the last inequality comes from Lemma C.3.

To summarize, we have that with probability at least $1 - 2\delta$, for every $i \in [K]$,

$$
\begin{aligned}
|\widehat{B}(i) - B(i)| &= \left| \frac{1}{K}\sum_{j=1}^K \left( \mu(\phi_{i,j}^\top \boldsymbol{\theta}^*) - \mu(\phi_{i,j}^\top \widehat{\boldsymbol{\theta}}) \right) \right| \\
&\leq \frac{1}{K}\sum_{j=1}^K \left| \mu(\phi_{i,j}^\top \boldsymbol{\theta}^*) - \mu(\phi_{i,j}^\top \widehat{\boldsymbol{\theta}}) \right| \\
&\leq \frac{L_\mu}{K}\sum_{j=1}^K \left| \phi_{i,j}^\top (\boldsymbol{\theta}^* - \widehat{\boldsymbol{\theta}}) \right| \\
&\leq \frac{3L_\mu\epsilon}{\kappa} \cdot \sqrt{\frac{d}{2}\log(1 + 2(\tau+N)/d) + \log(1/\delta)},
\end{aligned}
\tag{C.3}
$$

where the first equality is by the definition of the empirical/true Borda score; the first inequality is due to the triangle inequality; the second inequality is from the Lipschitz-ness of $\mu(\cdot)$ ($L_\mu = 1/4$ for the logistic function); the last inequality holds due to (C.2). This further implies the gap between the empirical Borda winner and the true Borda winner is bounded by:

$$
\begin{aligned}
B(i^*) - B(\widehat{i}) &= B(i^*) - \widehat{B}(i^*) + \widehat{B}(i^*) - B(\widehat{i}) \\
&\leq B(i^*) - \widehat{B}(i^*) + \widehat{B}(\widehat{i}) - B(\widehat{i}) \\
&\leq \frac{6L_\mu\epsilon}{\kappa} \cdot \sqrt{\frac{d}{2}\log(1 + 2(\tau+N)/d) + \log(1/\delta)},
\end{aligned}
$$

where the first inequality holds due to the definition of $\widehat{i}$, i.e., $\widehat{B}(\widehat{i}) \geq \widehat{B}(i)$ for any $i$; the last inequality holds due to (C.3).

Meanwhile, since $N := \sum_{(i,j)\in\text{supp}(\pi)} N(i,j)$ and $|\text{supp}(\pi)| \leq d(d+1)/2$ from Lemma C.1, we have that

$$
N \leq d(d+1)/2 + \frac{d}{\epsilon^2},
$$

because $\lceil x \rceil < x + 1$.

Therefore, with probability at least $1 - 2\delta$, the regret is bounded by:

$$\text{Regret}(T) = \text{Regret}_{1:\tau} + \text{Regret}_{\tau+1:\tau+N} + \text{Regret}_{\tau+N+1:T}$$

$$\leq \tau + N + \frac{12L_\mu \epsilon T}{\kappa} \cdot \sqrt{\frac{d}{2}\log(1 + 2(\tau + N)/d) + \log(1/\delta)}$$

$$\leq \tau + d(d+1)/2 + \frac{d}{\epsilon^2} + \frac{12L_\mu \epsilon T}{\kappa} \cdot O\left(d^{1/2}\sqrt{\log\left(\frac{T}{d\delta}\right)}\right)$$

$$= O\left(\kappa^{-1} d^{2/3} T^{2/3} \sqrt{\log\left(\frac{T}{d\delta}\right)}\right),$$

where the first equation is simply dividing the regret into 3 stages: 1 to $\tau$, $\tau + 1$ to $\tau + N$, and $\tau + N + 1$ to $T$; the second inequality is simply bounding the one-step regret from 1 to $\tau + N$ by 1, while for $t > \tau + N$, we have shown that the one-step regret is guaranteed to be smaller than $12L_\mu\epsilon\sqrt{d\log(1 + 2(\tau + N)/d) + \log(1/\delta)}/\sqrt{2}\kappa$. The last line holds because we set $\tau = O(d + \log(1/\delta))$ and $\epsilon = d^{1/6}T^{-1/3}$. Note that to ensure $\tau + N < T$, it suffices to assume $T = \Omega(d^2)$.

By setting $\delta = T^{-1}$, we can show that the expected regret of Algorithm 1 is bounded by

$$\widetilde{O}\left(\kappa^{-1}(d^{2/3}T^{2/3})\right).$$

$\square$

## C.2   PROOF OF THEOREM A.1

The following lemma characterizes the non-asymptotic behavior of the MLE estimator. It is extracted from Li et al. (2017).

**Lemma C.4** (Theorem 1, Li et al. 2017). *Define $\mathbf{V}_s = \sum_{t=1}^{s} \boldsymbol{\phi}_{i_t,j_t} \boldsymbol{\phi}_{i_t,j_t}^\top$, and $\widehat{\boldsymbol{\theta}}_s$ as the MLE estimator (5.1) in round $s$. If $\mathbf{V}_s$ satisfies*

$$\lambda_{\min}(\mathbf{V}_s) \geq \frac{512M_\mu^2(d^2 + \log(3/\delta))}{\kappa^4}, \tag{C.4}$$

*then for any fixed $\mathbf{x} \in \mathbb{R}^d$, with probability at least $1 - \delta$,*

$$|\langle \widehat{\boldsymbol{\theta}}_s - \boldsymbol{\theta}^*, \mathbf{x} \rangle| \leq \frac{3}{\kappa}\sqrt{\|\mathbf{x}\|_{\mathbf{V}_s^{-1}}^2 \log(3/\delta)}.$$

*Proof of Theorem A.1.* The proof can be essentially divided into three steps: 1. invoke Lemma C.2 to show that the initial $\tau$ rounds for exploration will guarantee (C.4) be satisfied; 2. invoke Lemma C.1 to obtain an optimal design $\pi$ and utilize Lemma C.4 to show that $|\langle \widehat{\boldsymbol{\theta}}_{\tau+N} - \boldsymbol{\theta}, \boldsymbol{\phi}_{i,j} \rangle| \leq 3\epsilon/\kappa$; 3. balance the not yet determined $\epsilon$ to obtain the regret upper bound.

First, we explain why we assume

$$T \geq \frac{C_3}{\kappa^6 \lambda_0^{3/2}} \max\left\{d^{5/2}, \frac{\log(K^2/\delta)}{\sqrt{d}}\right\}.$$

To ensure (C.4) in Lemma C.4 can hold, we resort to Lemma C.2, that is

$$\tau \geq \left(\frac{C_1\sqrt{d} + C_2\sqrt{\log(1/\delta)}}{\lambda_0}\right)^2 + \frac{2B}{\lambda_0},$$

$$B := \frac{512M_\mu^2(d^2 + \log(3/\delta))}{\kappa^4}.$$

Since we set $\tau = (d\log(K^2/\delta))^{1/3}T^{2/3}$, this means $T$ should be large enough, so that

$$(d\log(K^2/\delta))^{1/3}T^{2/3} \geq \left(\frac{C_1\sqrt{d} + C_2\sqrt{\log(1/\delta)}}{\lambda_0}\right)^2 + \frac{1024M_\mu^2(d^2 + \log(3K^2/\delta))}{\kappa^4\lambda_0}.$$

With a large enough universal constant $C_3$, it is easy to verify that the inequality above will hold as long as

$$T \geq \frac{C_3}{\kappa^6 \lambda_0^{3/2}} \max \left\{ d^{5/2}, \frac{\log(K^2/\delta)}{\sqrt{d}} \right\}.$$

By Lemma C.2, we have that with probability at least $1 - \delta$,

$$\lambda_{\min}(\mathbf{V}_\tau) \geq \frac{512 M_\mu^2 (d^2 + \log(3K^2/\delta))}{\kappa^4}. \tag{C.5}$$

From now on, we assume (C.5) always holds.

Define $N := \sum_{i,j} N(i,j)$, $\mathbf{V}_{\tau+1:\tau+N} := \sum_{t=\tau+1}^{\tau+N} \boldsymbol{\phi}_{i_t,j_t} \boldsymbol{\phi}_{i_t,j_t}^\top$, $\mathbf{V}_{\tau+N} := \mathbf{V}_\tau + \mathbf{V}_{\tau+1:\tau+N}$. Given the optimal design $\pi(i,j)$, the algorithm queries each pair $(i,j) \in \text{supp}(\pi)$ for exactly $N(i,j) = \lceil d\pi(i,j)/\epsilon^2 \rceil$ times. Therefore, the design matrix $\mathbf{V}_{\tau+N}$ satisfies

$$\begin{aligned}
\mathbf{V}_{\tau+N} &\succeq \mathbf{V}_{\tau+1:\tau+N} \\
&= \sum_{i,j} N(i,j) \boldsymbol{\phi}_{i,j} \boldsymbol{\phi}_{i,j}^\top \\
&\succeq \sum_{i,j} \frac{d\pi(i,j)}{\epsilon^2} \boldsymbol{\phi}_{i,j} \boldsymbol{\phi}_{i,j}^\top \\
&= \frac{d}{\epsilon^2} \mathbf{V}(\pi),
\end{aligned}$$

where $\mathbf{V}(\pi) := \sum_{i,j} \pi(i,j) \boldsymbol{\phi}_{i,j} \boldsymbol{\phi}_{i,j}^\top$. The first inequality holds because $\mathbf{V}_\tau$ is positive semi-definite, and the second inequality holds due to the choice of $N(i,j)$.

To invoke Lemma C.4, notice that $\lambda_{\min}(\mathbf{V}) \geq \lambda_{\min}(\mathbf{V}_\tau)$. Along with (C.5), by Lemma C.4, we have for any fixed $\boldsymbol{\phi}_{i,j}$, with probability at least $1 - \delta/K^2$, that

$$\begin{aligned}
|\langle \widehat{\boldsymbol{\theta}} - \boldsymbol{\theta}^*, \boldsymbol{\phi}_{i,j} \rangle| &\leq \frac{3}{\kappa} \sqrt{\|\boldsymbol{\phi}_{i,j}\|_{\mathbf{V}_{\tau+N}^{-1}}^2 \log(3K^2/\delta)} \\
&\leq \frac{3}{\kappa} \sqrt{\frac{\epsilon^2}{d} \cdot \|\boldsymbol{\phi}_{i,j}\|_{\mathbf{V}(\pi)^{-1}}^2 \log(3K^2/\delta)} \\
&= \frac{3\epsilon}{\kappa} \sqrt{\frac{\|\boldsymbol{\phi}_{i,j}\|_{\mathbf{V}(\pi)^{-1}}^2}{d}} \cdot \sqrt{\log(3K^2/\delta)} \\
&\leq \frac{3\epsilon}{\kappa} \cdot \sqrt{\log(3K^2/\delta)}, \tag{C.6}
\end{aligned}$$

where the first inequality comes from Lemma C.4; the second inequality holds because $\mathbf{V}_{\tau+N} \succeq \frac{d}{\epsilon^2} \mathbf{V}(\pi)$; the last inequality holds because $\pi$ is an optimal design and by Lemma C.1, $\|\boldsymbol{\phi}_{i,j}\|_{\mathbf{V}(\pi)^{-1}}^2 \leq d$.

Taking union bound for each $(i,j) \in [K] \times [K]$, we have that with probability at least $1 - \delta$, for every $i \in [K]$,

$$\begin{aligned}
|\widehat{B}(i) - B(i)| &= \left| \frac{1}{K} \sum_{j=1}^K \left( \mu(\boldsymbol{\phi}_{i,j}^\top \boldsymbol{\theta}^*) - \mu(\boldsymbol{\phi}_{i,j}^\top \widehat{\boldsymbol{\theta}}) \right) \right| \\
&\leq \frac{1}{K} \sum_{j=1}^K \left| \mu(\boldsymbol{\phi}_{i,j}^\top \boldsymbol{\theta}^*) - \mu(\boldsymbol{\phi}_{i,j}^\top \widehat{\boldsymbol{\theta}}) \right| \\
&\leq \frac{L_\mu}{K} \sum_{j=1}^K \left| \boldsymbol{\phi}_{i,j}^\top (\boldsymbol{\theta}^* - \widehat{\boldsymbol{\theta}}) \right| \\
&\leq \frac{3L_\mu \epsilon}{\kappa} \cdot \sqrt{\log(3K^2/\delta)}, \tag{C.7}
\end{aligned}$$

where the first equality is by the definition of the empirical/true Borda score; the first inequality is due to the triangle inequality; the second inequality is from the Lipschitz-ness of $\mu(\cdot)$ ($L_\mu = 1/4$ for the logistic function); the last inequality holds due to (C.6). This further implies the gap between the empirical Borda winner and the true Borda winner is bounded by:

$$
\begin{aligned}
B(i^*) - B(\widehat{i}) = B(i^*) - \widehat{B}(i^*) + \widehat{B}(i^*) - B(\widehat{i}) \\
\leq B(i^*) - \widehat{B}(i^*) + \widehat{B}(\widehat{i}) - B(\widehat{i}) \\
\leq \frac{6L_\mu \epsilon}{\kappa} \cdot \sqrt{\log(3K^2/\delta)},
\end{aligned}
$$

where the first inequality holds due to the definition of $\widehat{i}$, i.e., $\widehat{B}(\widehat{i}) \geq \widehat{B}(i)$ for any $i$; the last inequality holds due to (C.7).

Meanwhile, since $N := \sum_{(i,j) \in \mathrm{supp}(\pi)} N(i,j)$ and $|\mathrm{supp}(\pi)| \leq d(d+1)/2$ from Lemma C.1, we have that

$$
N \leq d(d+1)/2 + \frac{d}{\epsilon^2},
$$

because $\lceil x \rceil < x + 1$.

Therefore, with probability at least $1 - 2\delta$, the regret is bounded by:

$$
\begin{aligned}
\mathrm{Regret}(T) &= \mathrm{Regret}_{1:\tau} + \mathrm{Regret}_{\tau+1:\tau+N} + \mathrm{Regret}_{\tau+N+1:T} \\
&\leq \tau + N + \frac{12L_\mu \epsilon}{\kappa} T \cdot \sqrt{\log(3K^2/\delta)} \\
&\leq \tau + d(d+1)/2 + \frac{d}{\epsilon^2} + \frac{12L_\mu \epsilon}{\kappa} T \cdot \sqrt{\log(3K^2/\delta)} \\
&= O\big(\kappa^{-1}(d\log(K/\delta))^{1/3}T^{2/3}\big),
\end{aligned}
$$

where the first equation is simply dividing the regret into 3 stages: 1 to $\tau$, $\tau + 1$ to $\tau + N$, and $\tau + N + 1$ to $T$. the second inequality is simply bounding the one-step regret from 1 to $\tau + N$ by 1, while for $t > \tau + N$, we have shown that the one-step regret is guaranteed to be smaller than $12L_\mu \epsilon \sqrt{\log(3K^2/\delta)}/\kappa$. The last line holds because we set $\tau = (d\log(3K^2/\delta))^{1/3}T^{2/3}$ and $\epsilon = d^{1/3}T^{-1/3}\log(3K^2/\delta)^{-1/6}$.

By setting $\delta = T^{-1}$, we can show that the expected regret of Algorithm 1 is bounded by

$$
O\big(\kappa^{-1}(d\log(KT))^{1/3}T^{2/3}\big).
$$

Note that if there are exponentially many contextual vectors ($K \approx 2^d$), the upper bound becomes $\widetilde{O}(d^{2/3}T^{2/3})$. $\qquad\square$

## D    OMITTED PROOF IN SECTION 6

We make the following notation. Let $\mathcal{H}_{t-1} := (q_1, P_1, (i_1, j_1), r_1, \ldots, q_t, P_t)$ denotes the history up to time $t$. Here $P_t$ means the comparison probability $p_{i,j}^t$ in round $t$. The following lemmas are used in the proof. We first bound the estimate $\widehat{B}_t(i)$.

**Lemma D.1.** For all $t \in [T]$, $i \in [K]$, it holds that $\widehat{B}_t(i) \leq \lambda_0^{-1}/\gamma^2$.

*Proof of Lemma D.1.* Using our choice of $q_t \geq \gamma/K$, we have the following result for the matrix $Q_t$:

$$
Q_t = \sum_{i \in [K]} \sum_{j \in [K]} q_t(i)q_t(j)\phi_{i,j}\phi_{i,j}^\top \succeq \gamma^2 \frac{1}{K^2} \sum_{i \in [K]} \sum_{j \in [K]} \phi_{i,j}\phi_{i,j}^\top. \tag{D.1}
$$

Furthermore, we can use the definition of the estimate $\widehat{B}_t(i)$ to show that

$$
\begin{aligned}
\widehat{B}_t(i) = \left\langle \frac{1}{K} \sum_{j \in [K]} \phi_{i,j}, \widehat{\boldsymbol{\theta}}_t \right\rangle &= \left\langle \frac{1}{K} \sum_{j \in [K]} \phi_{i,j}, Q_t^{-1}\phi_{i_t,j_t} \right\rangle r_t(i_t, j_t) \\
&\leq \frac{1}{K} \sum_{j \in [K]} \|\phi_{i,j}\|_{Q_t^{-1}}^2,
\end{aligned}
$$

where we use the fact that $|r_t| \leq 1$. Let $\boldsymbol{\Sigma} = \frac{1}{K^2} \sum_{i=1}^{K} \sum_{j=1}^{K} \boldsymbol{\phi}_{i,j} \boldsymbol{\phi}_{i,j}^\top$. With (D.1) we have $Q_t \succeq \gamma^2 \boldsymbol{\Sigma}$. Therefore, we can further bound $\widehat{B}_t(i)$ with

$$\widehat{B}_t(i) \leq \frac{1}{K\gamma^2} \sum_{j \in [K]} \|\boldsymbol{\phi}_{i,j}\|_{\boldsymbol{\Sigma}^{-1}}^2$$

$$\leq \frac{1}{\gamma^2} \max_{i,j} \|\boldsymbol{\phi}_{i.j}\|_{\boldsymbol{\Sigma}^{-1}}^2$$

$$\leq \frac{\lambda_0^{-1}}{\gamma^2},$$

where the first inequality holds due to (D.1) and that $\|\mathbf{x}\|_{\mathbf{A}^{-1}}^2 \leq \|\mathbf{x}\|_{\mathbf{B}^{-1}}^2$ if $\mathbf{A} \succeq \mathbf{B}$; the third inequality holds because we assume $\lambda_0 \leq \lambda_{\min}\left(\frac{1}{K^2} \sum_{i=1}^{K} \sum_{j=1}^{K} \boldsymbol{\phi}_{i,j} \boldsymbol{\phi}_{i,j}^\top\right)$ and $\|\boldsymbol{\phi}_{i,j}\| \leq 1$. $\square$

The following lemma proves that our (shifted) estimate is unbiased.

**Lemma D.2.** For all $t \in [T]$, $i \in [K]$, the following equality holds:

$$\mathbb{E}[\widehat{B}_t(i)] = B_t(i) - \frac{1}{2}.$$

*Proof of Lemma D.2.* Using our definition of $\widehat{B}_t(i)$, we have

$$\widehat{B}_t(i) = \left\langle \frac{1}{K} \sum_{j \in [K]} \boldsymbol{\phi}_{i,j}, \widehat{\boldsymbol{\theta}}_t \right\rangle = \left\langle \frac{1}{K} \sum_{j \in [K]} \boldsymbol{\phi}_{i,j}, Q_t^{-1} \boldsymbol{\phi}_{i_t,j_t} \right\rangle r_t(i_t, j_t).$$

Therefore, by the law of total expectation (tower rule), we have

$$\mathbb{E}[\widehat{B}_t(i)] = \mathbb{E}_{\mathcal{H}_{t-1}}\left[\mathbb{E}_{(i_t,j_t,r_t)}\left[\left\langle \frac{1}{K} \sum_{j \in [K]} \boldsymbol{\phi}_{i,j}, Q_t^{-1} \boldsymbol{\phi}_{i_t,j_t} \right\rangle r_t(i_t, j_t) | \mathcal{H}_{t-1}\right]\right]$$

$$= \mathbb{E}_{\mathcal{H}_{t-1}}\left[\mathbb{E}_{(i_t,j_t)}\left[\left\langle \frac{1}{K} \sum_{j \in [K]} \boldsymbol{\phi}_{i,j}, Q_t^{-1} \boldsymbol{\phi}_{i_t,j_t} \right\rangle \mathbb{E}_{r_t}[r_t(i_t, j_t) | (i_t, j_t)] \Big| \mathcal{H}_{t-1}\right]\right]$$

$$= \mathbb{E}_{\mathcal{H}_{t-1}}\left[\mathbb{E}_{(i_t,j_t)}\left[\left\langle \frac{1}{K} \sum_{j \in [K]} \boldsymbol{\phi}_{i,j}, Q_t^{-1} \boldsymbol{\phi}_{i_t,j_t} \right\rangle p_t(i_t, j_t) \Big| \mathcal{H}_{t-1}\right]\right]$$

Then we use the definition of $p_t$ and the expectation. We can further get the equality

$$\mathbb{E}[\widehat{B}_t(i)] = \mathbb{E}_{\mathcal{H}_{t-1}}\left[\mathbb{E}_{(i_t,j_t)}\left[\left\langle \frac{1}{K} \sum_{j \in [K]} \boldsymbol{\phi}_{i,j}, Q_t^{-1} \boldsymbol{\phi}_{i_t,j_t} \boldsymbol{\phi}_{i_t,j_t}^\top \boldsymbol{\theta}^* \right\rangle \Big| \mathcal{H}_{t-1}\right]\right]$$

$$= \mathbb{E}_{\mathcal{H}_{t-1}}\left[\left\langle \frac{1}{K} \sum_{j \in [K]} \boldsymbol{\phi}_{i,j}, Q_t^{-1}\left(\sum_{i \in [K]} \sum_{j \in [K]} q_t(i) q_t(j) \boldsymbol{\phi}_{i,j} \boldsymbol{\phi}_{i,j}^\top\right) \boldsymbol{\theta}^* \right\rangle \Big| \mathcal{H}_{t-1}\right]$$

$$= \mathbb{E}_{\mathcal{H}_{t-1}}\left[\left\langle \frac{1}{K} \sum_{j \in [K]} \boldsymbol{\phi}_{i,j}, \boldsymbol{\theta}^* \right\rangle \Big| \mathcal{H}_{t-1}\right]$$

$$= B_t(i) - \frac{1}{2}.$$

Therefore, we have completed the proof of Lemma D.2. $\square$

The following lemma is similar to Lemma 5 in Saha et al. (2021b).

**Lemma D.3.** $\mathbb{E}_{\mathcal{H}_t}[q_t^\top \widehat{B}_t] = \mathbb{E}_{\mathcal{H}_{t-1}}\left[\mathbb{E}_{x \sim q_t}[B_t(x) | \mathcal{H}_{t-1}]\right] - \frac{1}{2}, \forall t \in [T]$.

*Proof of Lemma D.3.* Taking conditional expectation, we have

$$
\begin{aligned}
\mathbb{E}_{\mathcal{H}_t}[q_t^\top \widehat{B}_t] &= \mathbb{E}_{\mathcal{H}_t}\left[\sum_{i=1}^K q_t(i)\widehat{B}_t(i)\right] \\
&= \mathbb{E}_{\mathcal{H}_{t-1}}\left[\sum_{i=1}^K q_t(i)\mathbb{E}_{(i_t,j_t,r_t)}\left[\widehat{B}(i)\Big|\mathcal{H}_{t-1}\right]\right] \\
&= \mathbb{E}_{\mathcal{H}_{t-1}}\left[\sum_{i=1}^K q_t(i)\left(B_t(i)-\frac{1}{2}\right)\right] \\
&= \mathbb{E}_{\mathcal{H}_{t-1}}\left[\mathbb{E}_{x\sim q_t}\left[B_t(x)\Big|\mathcal{H}_{t-1}\right]\right] - \frac{1}{2},
\end{aligned}
$$

where we use the law of total expectation again as well as Lemma D.2. $\qquad\square$

The last lemma bounds a summation $\sum_{i\in[K]} q_t(i)\widehat{B}_t(i)^2$, which will be important in our proof.

**Lemma D.4.** At any time $t$, $\mathbb{E}[\sum_{i\in[K]} q_t(i)\widehat{B}_t(i)^2] \le d/\gamma$.

*Proof of Lemma D.4.* Let $\widehat{P}_t(i,j) = \langle \phi_{i,j}, \widehat{\theta}_t \rangle$. Using the definition of $\widehat{B}_t$ and $\widehat{P}_t(i,j)$, we have the following inequality:

$$
\begin{aligned}
\mathbb{E}\left[\sum_{i\in[K]} q_t(i)\widehat{B}_t(i)^2\right] &= \mathbb{E}\left[\sum_{i\in[K]} q_t(i)\left(\frac{1}{K}\sum_{j\in[K]}\widehat{P}_t(i,j)\right)^2\right] \\
&\le \mathbb{E}\left[\sum_{i\in[K]} q_t(i)\frac{1}{K}\sum_{j\in[K]}\widehat{P}_t^2(i,j)\right] \\
&= \mathbb{E}\left[\sum_{i\in[K]} q_t(i)\frac{1}{\gamma}\sum_{j\in[K]}\frac{\gamma}{K}\widehat{P}_t^2(i,j)\right] \\
&\le \frac{1}{\gamma}\mathbb{E}\left[\sum_{i\in[K]}\sum_{j\in[K]} q_t(i)q_t(j)\widehat{P}_t^2(i,j)\right].
\end{aligned}
$$

The first inequality holds due to the Cauchy-Schwartz inequality; the second inequality holds because the definition of $q_t$ satisfies $q_t \ge \gamma/K$.

Expanding the definition of $\widehat{P}_t^2(i,j)$, we have

$$
\begin{aligned}
\widehat{P}_t^2(i,j) &= r_t^2(i_t,j_t)\left(\phi_{i,j}^\top Q_t^{-1}\phi_{i_t,j_t}\right)^2 \\
&\le \phi_{i_t,j_t}^\top Q_t^{-1}\phi_{i,j}\phi_{i,j}^\top Q_t^{-1}\phi_{i_t,j_t},
\end{aligned}
$$

where we use $0 \le r_t^2(i_t,j_t) \le 1$. Therefore, the following inequality holds,

$$
\begin{aligned}
\sum_{i\in[K]}\sum_{j\in[K]} q_t(i)q_t(j)\widehat{P}_t^2(i,j) &\le \sum_{i\in[K]}\sum_{j\in[K]} q_t(i)q_t(j)\phi_{i_t,j_t}^\top Q_t^{-1}\phi_{i,j}\phi_{i,j}^\top Q_t^{-1}\phi_{i_t,j_t} \\
&= \phi_{i_t,j_t}^\top Q_t^{-1}\left(\sum_{i\in[K]}\sum_{j\in[K]} q_t(i)q_t(j)\phi_{i,j}\phi_{i,j}^\top\right)Q_t^{-1}\phi_{i_t,j_t} \\
&= \phi_{i_t,j_t}^\top Q_t^{-1}\phi_{i_t,j_t} \\
&= \text{trace}(\phi_{i_t,j_t}\phi_{i_t,j_t}^\top Q_t^{-1}).
\end{aligned}
$$

Using the property of trace, we have

$$
\mathbb{E}\left[\sum_{i\in[K]}\sum_{j\in[K]} q_t(i)q_t(j)\widehat{P}_t^2(i,j)\right] \le \text{trace}\left(\sum_{i\in[K]}\sum_{j\in[K]} q_t(i)q_t(j)\phi_{i,j}\phi_{i,j}^\top Q_t^{-1}\right) = d.
$$

Therefore, we finish the proof of Lemma D.4. $\qquad\square$

*Proof of Theorem 6.1.* Our regret is defined as follows,

$$
\mathbb{E}_{\mathcal{H}_T}[R_T] = \mathbb{E}_{\mathcal{H}_T}\left[\sum_{t=1}^{T}[2B_t(i^*) - B_t(i_t) - B_t(j_t)]\right]
$$

$$
= \max_{i\in[K]}\mathbb{E}_{\mathcal{H}_T}\left[\sum_{t=1}^{T}[2B_t(i) - B_t(i_t) - B_t(j_t)]\right].
$$

The second equality holds because $B_t$ and $i^*$ are independent of the randomness of the algorithm. Furthermore, we can write the expectation of the regret as

$$
\mathbb{E}_{\mathcal{H}_T}[R_T] = 2\max_{i\in[K]}\sum_{t=1}^{T}B_t(i) - \sum_{t=1}^{T}\mathbb{E}_{\mathcal{H}_T}[B_t(i_t) + B_t(j_t)]
$$

$$
= 2\max_{i\in[K]}\sum_{t=1}^{T}B_t(i) - 2\sum_{t=1}^{T}\mathbb{E}_{\mathcal{H}_{t-1}}[\mathbb{E}_{x\sim q_t}[B_t(x)|\mathcal{H}_{t-1}]]
$$

$$
= 2\max_{i\in[K]}\sum_{t=1}^{T}\left(B_t(i) - \frac{1}{2}\right) - 2\mathbb{E}_{\mathcal{H}_t}\left[q_t^\top \widehat{B}_t\right], \tag{D.2}
$$

where the last equality is due to Lemma D.3.

Then we follow the standard proof of EXP3 algorithm (Lattimore & Szepesvári, 2020). Let $S_{t,k} = \sum_{s=1}^{t}\left(B_s(k) - \frac{1}{2}\right)$, $\widehat{S}_{t,k} = \sum_{s=1}^{t}\widehat{B}_s(k)$, $\omega_t = \sum_{k\in[K]}\exp(-\eta\widehat{S}_{t,k})$ and $\omega_0 = K$. We have $\forall a \in [K]$,

$$
\exp(-\eta\widehat{S}_{T,a}) \le \sum_{k\in[K]}\exp(-\eta\widehat{S}_{T,k}) = \omega_T = \omega_0 \cdot \prod_{t=1}^{T}\frac{\omega_t}{\omega_{t-1}}. \tag{D.3}
$$

For each term in the product, we have

$$
\frac{\omega_t}{\omega_{t-1}} = \sum_{k\in[K]}\frac{\exp(-\eta\widehat{S}_{t-1,k})}{\omega_{t-1}} \cdot \exp(-\eta\widehat{B}_t(k))
$$

$$
= \sum_{k\in[K]}\widetilde{q}_t(k)\exp(-\eta\widehat{B}_t(k)), \tag{D.4}
$$

where the second equality holds because of the definition of $\widetilde{q}_t$. For any $\eta \le \lambda_0\gamma^2$, Lemma D.1 presents $|\eta\widehat{B}_t(k)| \le 1$. Thus, using the basic inequality $\exp(x) \le 1 + x + x^2/2$ when $x \le 1$, and $\exp(x) \ge 1 + x$, we have

$$
\frac{\omega_t}{\omega_{t-1}} \le \sum_{k\in[K]}\widetilde{q}_t(k)\left(1 - \eta\widehat{B}_t(k) + \eta^2\widehat{B}_t^2(k)\right)
$$

$$
= 1 - \eta\sum_{k\in[K]}\widetilde{q}_t(k)\widehat{B}_t(k) + \eta^2\sum_{k\in[K]}\widetilde{q}_t(k)\widehat{B}_t^2(k)
$$

$$
\le \exp\left(-\eta\sum_{k\in[K]}\widetilde{q}_t(k)\widehat{B}_t(k) + \eta^2\sum_{k\in[K]}\widetilde{q}_t(k)\widehat{B}_t^2(k)\right). \tag{D.5}
$$

Combining (D.3), (D.4) and (D.5), we have

$$\exp(-\eta \widehat{S}_{T,a}) \leq K \exp\left(\sum_{t=1}^{T}\left[-\eta \sum_{k \in [K]} \widetilde{q}_t(k)\widehat{B}_t(k) + \eta^2 \sum_{k \in [K]} \widetilde{q}_t(k)\widehat{B}_t^2(k)\right]\right),$$

and therefore

$$\sum_{t=1}^{T} \widehat{B}_t(a) - \sum_{t=1}^{T} \widetilde{q}_t^\top \widehat{B}_t \leq \frac{\log K}{\eta} + \eta \sum_{t=1}^{T} \sum_{k \in [K]} \widetilde{q}_t(k)\widehat{B}_t^2(k).$$

Since $\widetilde{q}_t = \frac{q_t - \gamma/K}{1-\gamma}$, we have

$$(1-\gamma)\sum_{t=1}^{T} \widehat{B}_t(a) - \sum_{t=1}^{T} q_t^\top \widehat{B}_t \leq \frac{\log K}{\eta} + \eta \sum_{t=1}^{T} \sum_{k \in [K]} \widetilde{q}_t(k)\widehat{B}_t^2(k).$$

Choosing $a = i^*$, changing the summation index to $i$ and taking expectation on both sides, we have

$$(1-\gamma)\mathbb{E}_{\mathcal{H}_T}\sum_{t=1}^{T} \widehat{B}_t(i^*) - \sum_{t=1}^{T} \mathbb{E}_{\mathcal{H}_T}\left[q_t^\top \widehat{B}_t\right] \leq \frac{\log K}{\eta} + \mathbb{E}_{\mathcal{H}_T}\left[\eta \sum_{t=1}^{T} \sum_{i \in [K]} q_t(i)\widehat{B}_t^2(i)\right].$$

Substituting the above inequality into (D.2) and using Lemma D.2, D.3, we can bound the regret as

$$\begin{aligned}
\mathbb{E}[R_T] &\leq 2\gamma T + \frac{2\log K}{\eta} + 2\eta \sum_{t=1}^{T} \mathbb{E}_{\mathcal{H}_T}\left[\sum_{i \in [K]} q_t(i)s_t(i)^2\right] \\
&\leq 2\gamma T + 2\frac{\log K}{\eta} + \frac{2\eta dT}{\gamma} \\
&\leq 2(\log K)^{1/3} d^{1/3} T^{2/3} \sqrt{1/\lambda_0} + 2(\log K)^{1/3} d^{1/3} T^{2/3} + 2(\log K)^{1/3} d^{1/3} T^{2/3} \sqrt{\lambda_0},
\end{aligned}$$

where the second inequality holds due to Lemma D.4. In the last inequality, we put in our choice of parameters $\eta = (\log K)^{2/3} d^{-1/3} T^{-2/3}$ and $\gamma = \sqrt{\eta d/\lambda_0} = (\log K)^{1/3} d^{1/3} T^{-1/3} \lambda_0^{-1/2}$. This finishes our proof of Theorem 6.1. $\qquad\square$

# E   DETAILED EXPLANATION OF STUDIED ALGORITHMS IN EXPERIMENTS

The following list summarizes all methods we implemented:

BETC-GLM(-MATCH): Algorithm 1 proposed in this paper. For general link function, to find $\widehat{\theta}$ by MLE in (5.1), 100 rounds of gradient descent are performed. The failure probability is set to $\delta = 1/T$. Parameters $\tau$ and $\epsilon$ are set to values listed in Theorem A.1. For BETC-GLM-MATCH, we use the $\tau$ and $\epsilon$ outlined in Theorem 5.1.

UCB-BORDA: The UCB algorithm (Auer et al., 2002) using *Borda reduction* technique mentioned by Busa-Fekete et al. (2018). The complete listing is displayed in Algorithm 3.

DEXP3: Dueling-Exp3 is an adversarial Borda bandit algorithm developed by Saha et al. (2021a), which also applies to our stationary bandit case. Relevant tuning parameters are set according to their upper-bound proof.

ETC-BORDA: We devise a simple explore-then-commit algorithm, named ETC-BORDA. Like DEXP3, ETC-BORDA does not take any contextual information into account. The complete procedure of ETC-BORDA is displayed in Algorithm 4, Appendix G.2. The failure probability $\delta$ is optimized as $1/T$.

BEXP3: The proposed method for adversarial Borda bandits displayed in Algorithm 2. $\eta$ and $\gamma$ are chosen to be the value stated in Theorem 6.1.

# F   REAL-WORLD DATA EXPERIMENTS

To showcase the performance of the algorithms in a real-world setting, we use EventTime dataset (Zhang et al., 2016). In this dataset, $K = 100$ historical events are compared in a pairwise fashion by crowd-sourced workers.

We first calculate the empirical preference probabilities $\widetilde{p}_{i,j}$ from the collected responses. A visualized preferential matrix consisting of $\widetilde{p}_{i,j}$ is shown in Figure 5 in Appendix F.1, which demonstrates that STI and SST conditions hardly hold in reality. During simulation, $\widetilde{p}_{i,j}$ is the parameter of the Bernoulli distribution that is used to generate the responses whenever a pair $(i, j)$ is queried. The contextual vectors $\phi_{i,j}$ are generated randomly from $\{-1, +1\}^5$. For simplicity, we assign the item pairs that have the same probability value with the same contextual vector, i.e., if $\widetilde{p}_{i,j} = \widetilde{p}_{k,l}$ then $\phi_{i,j} = \phi_{k,l}$. The MLE estimator $\widehat{\theta}$ in (5.1) is obtained to construct the recovered preference probability $\widehat{p}_{i,j} := \mu(\phi_{i,j}^\top \widehat{\theta})$ where $\mu(x) = 1/(1 + e^{-x})$ is the logistic function. We ensure that the recovered preference probability $\widehat{p}_{i,j}$ is close to $\widetilde{p}_{i,j}$, so that $\phi_{i,j}$ are informative enough. As shown in Figure 3, our algorithm outperforms the baseline methods as expected. In particular, the gap between our algorithm and the baselines is even larger than that under the generated hard case. In both settings, our algorithms demonstrated a stable performance with negligible variance.

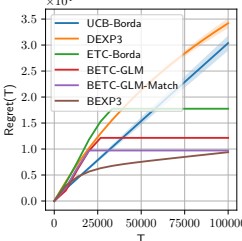

Figure 3: EventTime

Figure 4: The regret of the proposed algorithm (BETC-GLM,BEXP3) and the baseline algorithms (UCB-BORDA, DEXP3, ETC-BORDA).

### F.1 DATA VISUALIZATION

The events in EventTime dataset are ordered by the time they occurred. In Figure 5, the magnitude of each $\widetilde{p}_{i,j}$ is color coded. It is apparent that there is no total/consistent ordering (i.e., $\widetilde{p}_{i,j} > \frac{1}{2} \Leftrightarrow i \succ j$) can be inferred from this matrix due to inconsistencies in the ordering and many potential paradoxes. Hence STI and SST can hardly hold in this case.

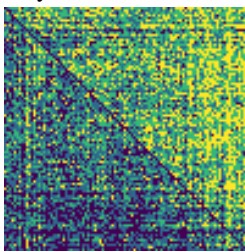

Figure 5: Estimated preferential matrix consists of $\widetilde{p}_{i,j}$ from the EventTime dataset.

## G ADDITIONAL INFORMATION FOR EXPERIMENTS

### G.1 THE UCB-BORDA ALGORITHM

The UCB-BORDA procedure, displayed in Algorithm 3 is a UCB algorithm with Borda reduction only capable of minimization of regret in the following form:

$$\text{Regret}(T) = \sum_{t=1}^{T} \big(B(i^*) - B(i_t)\big).$$

Let $\mathbf{n}_i$ be the number of times arm $i \in [K]$ has been queried. Let $\mathbf{w}_i$ be the number of times arm $i$ wins the duel. $\widehat{B}(i)$ is the estimated Borda score. $\alpha$ is set to 0.3 in all experiments.

---
**Algorithm 3** UCB-BORDA
---
1: **Input:** time horizon $T$, number of items $K$, exploration parameter $\alpha$.
2: **Initialize:** $\mathbf{n} = \mathbf{w} = \{0\}^K$, $\widehat{B}(i) = \frac{1}{2}, i \in [K]$
3: **for** $t = 1, 2, \ldots, T$ **do**
4:     $i_t = \text{argmax}_{k \in [K]} \big(\widehat{B}_k + \sqrt{\frac{\alpha \log(t)}{\mathbf{n}_k}}\big)$
5:     sample $j_t \sim \text{Uniform}([K])$
6:     query pair $(i_t, j_t)$ and receive feedback $r_t \sim \text{Bernoulli}(p_{i_t, j_t})$
7:     $\mathbf{n}_{i_t} = \mathbf{n}_{i_t} + 1, \mathbf{w}_{i_t} = \mathbf{w}_{i_t} + r_t, \widehat{B}(i_t) = \frac{\mathbf{w}_{i_t}}{\mathbf{n}_{i_t}}$
8: **end for**
---

### G.2 THE ETC-BORDA ALGORITHM

The ETC-BORDA procedure, displayed in Algorithm 4 is an explore-then-commit type algorithm capable of minimizing the Borda dueling regret. It can be shown that the regret of Algorithm 4 is $\widetilde{O}(K^{1/3}T^{2/3})$.

---

**Algorithm 4** ETC-BORDA

1: **Input:** time horizon $T$, number of items $K$, target failure probability $\delta$
2: **Initialize:** $\mathbf{n} = \mathbf{w} = \{0\}^K$, $\widehat{B}(i) = \frac{1}{2}, i \in [K]$
3: Set $N = \lceil K^{-2/3} T^{2/3} \log(K/\delta)^{1/3} \rceil$
4: **for** $t = 1, 2, \ldots, T$ **do**
5:     Choose action $i_t \leftarrow \begin{cases} 1 + (t-1) \bmod K, & \text{if } t \leq KN, \\ \text{argmax}_{i \in [K]} \widehat{B}(i), & \text{if } t > KN. \end{cases}$
6:     Choose action $j_t = \begin{cases} \text{Uniform}([K]), & \text{if } t \leq KN, \\ \text{argmax}_{i \in [K]} \widehat{B}(i), & \text{if } t > KN. \end{cases}$
7:     query pair $(i_t, j_t)$ and receive feedback $r_t \sim \text{Bernoulli}(p_{i_t, j_t})$
8:     **if** $t \leq N$ **then**
9:         $\mathbf{n}_{i_t} = \mathbf{n}_{i_t} + 1, \mathbf{w}_{i_t} = \mathbf{w}_{i_t} + r_t, \widehat{B}(i_t) = \frac{\mathbf{w}_{i_t}}{\mathbf{n}_{i_t}}$
10:    **end if**
11: **end for**

---

### G.3 FRANK-WOLFE ALGORITHM USED TO FIND APPROXIMATE SOLUTION FOR G-OPTIMAL DESIGN

In order to find a solution for the G-optimal design problem, we resort to the Frank-Wolfe algorithm to find an approximate solution. The detailed procedure is listed in Algorithm 5. In Line 4, each outer product costs $d^2$ multiplications, $K^2$ such matrices are scaled and summed into a $d$-by-$d$ matrix $\mathbf{V}(\pi)$, which costs $O(K^2 d^2)$ operations in total. In Line 5, one matrix inversion costs approximately $O(d^3)$. The weighted norm requires $O(d^2)$ and the maximum is taken over $K^2$ such calculated values. The scaling and update in the following lines only require $O(K^2)$. In summary, the algorithm is dominated by the calculation in Line 5 which costs $O(d^2 K^2)$.

In experiments, the G-optimal design $\pi(i, j)$ is approximated by running 20 iterations of Frank-Wolfe algorithm, which is more than enough for its convergence given our particular problem instance. (See Note 21.2 in (Lattimore & Szepesvári, 2020)).

---

**Algorithm 5** G-OPTIMAL DESIGN BY FRANK-WOLFE

1: **Input:** number of items $K$, contextual vectors $\boldsymbol{\phi}_{i,j}, i \in [K], j \in [K]$, number of iterations $R$
2: **Initialize:** $\pi_1(i, j) = 1/K^2$
3: **for** $r = 1, 2, \cdots, R$ **do**
4:     $\mathbf{V}(\pi_r) = \sum_{i,j} \pi_r(i, j) \boldsymbol{\phi}_{i,j} \boldsymbol{\phi}_{i,j}^\top$
5:     $i_r^*, j_r^* = \text{argmax}_{(i,j) \in [K] \times [K]} \|\boldsymbol{\phi}_{i,j}\|_{\mathbf{V}(\pi_r)^{-1}}$
6:     $g_r = \|\boldsymbol{\phi}_{i_r^*, j_r^*}\|_{\mathbf{V}(\pi_r)^{-1}}$
7:     $\gamma_r = \frac{g_r - 1/d}{g_r - 1}$
8:     $\pi_{r+1}(i, j) = (1 - \gamma_r) \pi_r(i, j) + \gamma_r \mathbb{1}(i_r^* = i) \mathbb{1}(j_r^* = j)$
9: **end for**
10: **Output:** Approximate G-optimal design solution $\pi_{R+1}(i, j)$

---

# H BEXP3 WITH MANY ARMS

In this section, we discuss how our algorithm BEXP3 can be modified to obtain a $\widetilde{O}(d^{2/3}T^{2/3})$ regret, when the number of arms $K$ is exponentially large ($\log K = \omega(d)$).

The idea is to first construct an $\epsilon$-cover $\mathcal{S}$ over the set of mean vectors $\left\{ \frac{1}{K} \sum_{j \in [K]} \phi_{i,j} \right\}_{i \in [K]}$, so that for any $i \in [K]$, there exist a vector $\psi_i \in \mathcal{S}$ such that $\left\| \frac{1}{K} \sum_{j \in [K]} \phi_{i,j} - \psi_i \right\|_2 \leq \epsilon$. In the $d$-dimensional space, it suffices to choose $|\mathcal{S}| = O((1/\epsilon)^d)$ representative vectors to cover all the vectors.

---
**Algorithm 6** BEXP3 with large $K$

---
1: **Input:** time horizon $T$, number of items $K$, feature dimension $d$, feature vectors $\phi_{i,j}$ for $i \in [K]$, $j \in [K]$, learning rate $\eta$, exploration parameter $\gamma$, covering radius $\epsilon$.
2: **Initialize:** $q_1(i) = \frac{1}{K}$.
3: Calculate an $\epsilon$-cover $\mathcal{S}$ of $\left\{ \frac{1}{K} \sum_{j \in [K]} \phi_{i,j} \right\}_{i \in [K]}$, with $|\mathcal{S}| = O((1/\epsilon)^d)$.
4: Denote the vector in $\mathcal{S}$ closest to $\frac{1}{K} \sum_{j \in [K]} \phi_{i,j}$ as $\psi_i$ for all $i \in [K]$.
5: **for** $t = 1, \ldots, T$ **do**
6:    Sample items $i_t \sim q_t$, $j_t \sim q_t$.
7:    Query pair $(i_t, j_t)$ and receive feedback $r_t$ .
8:    Calculate $Q_t = \sum_{i \in [K]} \sum_{j \in [K]} q_t(i) q_t(j) \phi_{i,j} \phi_{i,j}^\top$, $\widehat{\theta}_t = Q_t^{-1} \phi_{i_t,j_t} r_t$.
9:    Calculate the (shifted) Borda score estimates $\widehat{B}_t(i) = \langle \psi_i, \widehat{\theta}_t \rangle$.
10:   Update for all $i \in [K]$, set

$$\widetilde{q}_{t+1}(i) = \frac{\exp(\eta \sum_{l=1}^t \widehat{B}_l(i))}{\sum_{j \in [K]} \exp(\eta \sum_{l=1}^t \widehat{B}_l(j))}; \qquad q_{t+1}(i) = (1-\gamma)\widetilde{q}_{t+1}(i) + \frac{\gamma}{K}.$$

11: **end for**

---

Now BEXP3 will be performed as the original case except that at Line 9 in Algorithm 6, we replace the average contextual vectors $\frac{1}{K} \sum_{j=1}^K \phi_{i,j}$ with those nearest representative vectors $\phi_i$. Since there are only $|\mathcal{S}| = O((1/\epsilon)^d)$ unique rewards, the EXP3 argument (D.3) can be performed on $|\mathcal{S}|$ unique arms instead of $K$ arms:

$$\exp(-\eta \widehat{S}_{T,a}) \leq \sum_{\phi \in \mathcal{S}} \exp(-\eta \widehat{S}_{T,\phi}) = \omega_T = \omega_0 \cdot \prod_{t=1}^T \frac{\omega_t}{\omega_{t-1}},$$

where $\omega_0 = |\mathcal{S}|$ instead of $K$.

Eventually the algorithm suffers a regret of $\widetilde{O}(d^{2/3}T^{2/3} \log^{1/3}(1/\epsilon))$. The $\epsilon$-net incurs an additional approximation error of order $O(\epsilon T)$. Setting $\epsilon = T^{-1}$ will improve the regret to $d^{2/3}T^{2/3}$ up to log factors.

# I ADDITIONAL COMPARISON BETWEEN BETC AND BEXP3

Here we show under the same experimental setting, tuning the error tolerance $\epsilon$ in BETC can further reduce its total regret up to a constant factor, showing that under suitable hyper-parameter choices, BETC can outperform BEXP3.

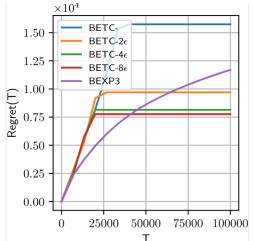

Figure 6: The performance of BETC under different choices of error tolerance $\epsilon$, compared with BEXP3. We examined BETC with $\epsilon, 2\epsilon, 4\epsilon, 8\epsilon$ where $\epsilon = d^{1/6}T^{-1/3}$.

