# OpenReview forum: "Borda Regret Minimization for Generalized Linear Dueling Bandits"
_ICLR.cc/2024/Conference — Submitted to ICLR 2024_

### Official Review · Reviewer_Ln8Y · 2023-10-22

**Soundness:** 3 good
**Presentation:** 3 good
**Contribution:** 3 good
**Rating:** 6
**Confidence:** 3

**Summary:**

In this work, the authors consider the problem of linear dueling bandits, both in the stochastic and the adversarial setting. Dueling bandits is a variant of the standard multi-armed bandits problem, particularly well suited for practical applications, where at each round the algorithm selects two arms (also called items) to play and receives receives feedback that expresses which of the two items was preferred, As this preferences don't necessarily imply the existence of a unique ranking between the arms, the authors of the papers consider the Borda score (which is the average win rate against all arms) as the measure used to define the regret.
The goal of this work is to take advantage of the side information given by the linear bandit structure to learn faster.

The first contribution of the paper is a well detailed lower bound proof, which shows that the difficulty of the problem scales as $d^{2/3}T^{2/3}$, where $T$  is the time horizon and $d$ is the dimension of the contextual vectors.
They then present a simple Explore then commit strategy that achieves near optimal performance for the stochastic version of the problem, and a variation of the EXP3 algorithm that also achieves near optimal (and optimal rate when the number of arms is of order $2^d$).
Finally, they present experiments on both generated and real world datasets showing that the presented algorithms can successfully use the linear structure of the problem to outperform the all the baselines.

**Strengths:**

This paper considers a new variation of the linear dueling bandits problem using Borda regret and provide very complete results. Specifically, the most important result presented in this paper is the lower bound, which is well detailed.

This first result is crucial to justify that the proposed algorithms, that shine by their simplicity, are sufficient for the problem at hand. I find the dual explore structure of the ETC algorithm fairly interesting: in the first part, the exploration is uniform to initialize correctly the algorithm, but then the exploration is refined to be tuned according to the precision required for each pairs of arms. The authors derive near optimal high probability bounds for this algorithm, which are stronger guarantees compared to results holding in expectation.

The second algorithm builds upon the very standard EXP3 algorithm, and its DEXP3 variant for dueling bandits. This algorithm is more robust, as it holds for the adversarial version of the problem, which is more challenging. Interestingly, the proposed bound is actually tight when the number of arms is exponential in the dimension of the contextual vector, and in the experiments, BEXP3 outperforms its ETC counterpart.

It is also appreciated that the authors discuss how to modify the algorithms to adapt for small number of arms as well as infinite number of arm, and it is worth noting that the paper is particularly well written, with details like ensuring that all notations are clearly defined and visual representations of the lower bound.

**Weaknesses:**

This work seems very solid overall.

One limitation that seems perhaps unnecessary is the fact that the time horizon has to be known. It would be nice for the authors to discuss which approaches (such as a doubling trick) could be used to remove this limitation, which would make these algorithms even easier to use in practice. (For the EXP3 algorithm, it is now standard to use a time varying learning rate for more applications).

In the experiments section, it would be nice to see more experiments that compare the two presented algorithms with different number of arms and time horizons: Will the ETC algorithm always have larger regret than the BEXP3 one due to the cost of the exploration phase or are there problem settings in which the ETC algorithm is better? as the BEXP3 algorithm is more robust and performs better in the experiments, I wonder if there is any case where one would prefer the ETC algorithm?

**Questions:**

Besides for the original version of the EXP3 algorithm (Auer et al. 2002), it is more common to find this algorithm stated with the use of losses rather than rewards, as it is not necessary to include extra exploration. Have you considered converting these rewards into losses and relying on EXP3 with losses (and possibly with time dependent leaning rate?)

(as stated in the weaknesses:) Will the ETC algorithm always have larger regret than the BEXP3 one due to the cost of the exploration phase or are there problem settings in which the ETC algorithm is better? as the BEXP3 algorithm is more robust and performs better in the experiments, I wonder if there is any case where one would prefer the ETC algorithm?

---

> ### Author Response · Authors · 2023-11-17
> **Author Response to Reviewer Ln8Y**
>
> Thank you for your positive feedback and strong support. We address your comments and questions as follows:
>
> ---
>
> **Q1**: Besides for the original version of the EXP3 algorithm (Auer et al. 2002), it is more common to find this algorithm stated with the use of losses rather than rewards, as it is not necessary to include extra exploration. Have you considered converting these rewards into losses and relying on EXP3 with losses (and possibly with time-dependent leaning rate?) \
> **A1**: We believe the reward formulation and loss formulation are essentially equivalent to a negation sign. For an unknown time horizon, indeed, we can incorporate a time-varying learning rate. For simplicity, we did not add this variant. We will add comments on this.
>
> ---
>
> **Q2**: Will the ETC algorithm always have larger regret than the BEXP3 one due to the cost of the exploration phase or are there problem settings in which the ETC algorithm is better? as the BEXP3 algorithm is more robust and performs better in the experiments, I wonder if there is any case where one would prefer the ETC algorithm? \
> **A2**: *ETC versus EXP3*: Two algorithms have essentially the same order of regret in our setting because $K = 2^d$. During our experiment, we did not carefully tune the constant in front of each parameter. As shown in Appendix I of the revised version, tuning the parameters of BETC can lead to an improvement of up to 50%. Our belief is that two algorithms should perform equally well in a stochastic setting, and only differ by a constant order.
>
> *When can BETC beat BEXP3?* BETC can work for general link functions, while BEXP3 is guaranteed to work only for the linear model. Even though in our experiments BEXP3 worked well with the logistic link function, it is not guaranteed to work well under any link function. For a less favorable link function, it might fail.\
> Besides, our current implementation of BEXP3 requires $K$ to be not too large $K = O(2^d)$, which is often true in real-world applications. When $K$ is extremely large, then BETC is preferred as the sample complexity does not depend on $K$ but only on the dimension $d$. (see Q2 in our response to Reviewer Kuh2 for more details)

---

> > ### Comment · Reviewer_Ln8Y · 2023-11-22
> >
> > Thank you for the added details

---

> > > ### Comment · Reviewer_Ln8Y · 2023-11-22
> > >
> > > After reading your rebuttal and the considering the other reviewers concerns, I have adjusted my score. I believe that the results are still interesting, but the novelty is indeed not as high as I had previously assumed.

---

### Official Review · Reviewer_PcGb · 2023-10-29

**Soundness:** 2 fair
**Presentation:** 3 good
**Contribution:** 2 fair
**Rating:** 5
**Confidence:** 2

**Summary:**

This paper delves into the realm of generalized linear dueling bandits within a stochastic setting and linear bandits within an adversarial setting. The scenario involves K arms with fixed features, and at each time step, the agent selects a pair of items (i_t, j_t), receiving stochastic feedback indicating whether i_t is preferred over item j_t. The probability model adopted encompasses a generalized linear model with an unknown parameter $\theta^*\in\mathbb{R}^d$ for the stochastic setting and linear model with $\theta_t$ for the adversarial setting. Regret is assessed through the Borda score, defined as the average winning probability of an arm over the other arms.

The authors establish a lower bound for both the stochastic and adversarial settings. For the stochastic setting, they introduce an algorithm based on ETC, tightly matching the lower bound concerning T and d. In the adversarial setting, they propose the BEXP3 algorithm based on EXP3, achieving regret of (dlog K)^{1/3}T^{2/3}. The paper concludes with a demonstration of the proposed algorithms using synthetic and real-world datasets.

**Strengths:**

The authors explore generalized linear dueling bandits with Borda scores, conducting an analysis of regret lower bounds and presenting algorithms for both stochastic and adversarial settings with upper bounds on regret.

**Weaknesses:**

-The authors assert that previous work by Saha (2021) on linear contextual duel bandits can be considered a special case of their model. However, the cited work involves a contextual set of arms that may change over time and a more generalized Multi-nomial logistic model, in contrast to the fixed feature vectors and dueling bandits considered in this study. Notably, the previously proposed algorithm achieves a regret bound of $\sqrt{T}$, while the algorithm presented in this work achieves a $T^{2/3}$ regret bound.

-As acknowledged by the authors, Saha (2021a) addressed adversarial duel bandits. While this present study introduces a linear model for the adversarial case, the extension to a linear model appears to follow the adversarial linear bandit algorithm outlined in "Bandit Algorithms" by Lattimore and Szepesvari and D-EXP3 [Saha 2021a]. There is a concern regarding whether there are discernible factors indicating that this extension is not a trivial one.

**Questions:**

-As indicated in the Weakness section, what accounts for the fact that the previously proposed algorithm in [Saha 2021] attains a regret bound of $\sqrt{T}$—which appears to be superior to the results presented in this work, specifically $T^{2/3}$ in both lower and upper bounds?

-Regarding Theorem 4.1, what constitutes the primary technical challenge in analyzing the lower bound for the linear bandit model compared to the Multi-Armed Bandit (MAB) scenario discussed in [Saha 2021a]?

---

> ### Author Response · Authors · 2023-11-17
> **Author Response to Reviewer PcGb**
>
> Thank you for your feedback! We address your comments and questions as follows:
>
> ---
>
> **Q1**: The authors assert that previous work by Saha (2021) on linear contextual duel bandits can be considered a special case of their model. However, the cited work involves a contextual set of arms that may change over time and a more generalized Multi-nomial logistic model, in contrast to the fixed feature vectors and dueling bandits considered in this study. \
> **A1**: As explained in Section A.1, in Saha (2021), the assumption is that each item has a `score’ $v_i = x_i^{\top} \theta^*$. Then, a changing set of arms is possible as long as the comparison follows the logistic model $P(i \succ j) = \mu(v_i - v_j) = \mu((x_i - x_j)^{\top}\theta^*)$. Similarly, the generalized multi-nomial logistic model can be satisfied as $P(i \text{ chosen among all items}) \propto e^{v_i}$.
>
> We claimed that our setting covers theirs because we consider $P(i \succ j) = \mu(\phi_{i,j}^{\top} \theta^*)$, and if $\phi_{i,j} = x_i - x_j$, then our setting becomes theirs, albeit with fixed arms and pairwise comparison only. On the other hand, in our general setting, $x_i$ is not available. Then it will not allow either a set of arms that may change over time or a multi-nomial logistic model that depends on the existence of $x_i$.
>
> ---
>
> **Q2**:   Why the previously proposed algorithm in [Saha 2021] attain a regret bound of  $\sqrt{T}$ which appears to be superior to the results presented in this work, specifically $T^{2/3}$ in both lower and upper bounds? \
> **A2**: We believe there is some misunderstanding here. As explained in the last question if we do only have $\phi_{i,j}$ instead of $x_i$ and $x_j$, then the algorithms in [Saha 2021] cannot work at all in our setting, because they have no access to the non-existent $x_i$, not to mention a $\sqrt{T}$ regret.
>
> We can only achieve $T^{2/3}$ regret because our studied problem is a more general one, as explained above. And a more general class of problems will incur no less regret than a subset of problems does. So it is normal to have their $\sqrt{T}$ regret for a subset of problems against our $T^{2/3}$ for the whole problem set.
>
> ---
>
> **Q3**: Regarding Theorem 4.1, what constitutes the primary technical challenge in analyzing the lower bound for the linear bandit model compared to the Multi-Armed Bandit (MAB) scenario discussed in [Saha 2021a]? \
> **A3**:  In the previous work [Saha et al 21], there are $K-1$ good arms and only the best one of them differs from the others. This design will not work in our setting as it leads to lower bounds sub-optimal in dimension $d$. The new construction of our lower bound is based on the hardness of identifying the best arm in the $d$-dimensional linear bandit setting, which is quite different from the multi-armed setting.
>
> [Saha et al 21]’s proof is directly based on hypothesis testing: either identifying the best arm with gap $\epsilon$ within $T$ rounds (if $T > \frac{K}{1440 \epsilon^3}$) or incurring $\epsilon T$ regret (if $T \le \frac{K}{1440 \epsilon^3}$). In contrast, our proof technique bounds from below the regret by the expected number of sub-optimal arm pulls and does not divide the problem instances into two cases (i.e. whether $T≤\frac{K}{1440 \epsilon^3}$). To prove the lower bound, we first apply a new reduction step to restrict the choice of $i_t$. Then we bound from below the regret by the expected number of sub-optimal arm pulls.

---

> > ### Comment · Reviewer_PcGb · 2023-11-20
> >
> > I appreciate your clarification regarding the distinction between the current work and the prior study by Saha in 2021.
> >
> > Aligning with the comments by other reviewers, I agree that adopting Borda regret simplifies the problem, and the suggested algorithms seem to be a straightforward extension of earlier work (Saha et al. 2021a, "Bandit Algorithms" by Lattimore and Szepesvari). Consequently, I will keep my original score.

---

> > > ### Author Response · Authors · 2023-11-22
> > >
> > > We appreciate your further feedback.
> > >
> > > We believe there is a misjudgment by Reviewer vgjv. The claim that adopting Borda regret simplifies the problem is not well grounded. Note that in our setting, there is no coherent ranking, thus finding the Borda winner is a natural and meaningful learning objective.
> > >
> > > As for your concerns regarding our contribution compared with the results in dueling bandits with a finite number of arms (Saha 2021a), we have explained that in our previous response. We feel that using existing results of multi-armed dueling bandits to diminish the significance of contextual dueling bandits is not fair, especially considering our work is the first work on contextual dueling bandits without a coherent ranking.

---

### Official Review · Reviewer_Kuh2 · 2023-10-31

**Soundness:** 3 good
**Presentation:** 3 good
**Contribution:** 2 fair
**Rating:** 5
**Confidence:** 4

**Summary:**

The paper studies generalized linear dueling bandits with the goal of tracking the Borda winner and minimizing Borda regret. They have matching upper and lower gap-free regret bounds in the stochastic setting and a regret upper bound in the adversarial setting.

While I believe the proofs are technically correct, I think the results are not surprising and so the novelty is limited. Overall, this paper seems to mostly combine two well-known theories in a straightforward application: (1) that of Borda regret minimization in dueling bandits (especially calling on the results of Saha et al., 2021) and (2) known techniques for generalized linear bandits (e.g., Li et al. 2017). Plese see "Weaknesses" below for specific discussions.

**Strengths:**

* I appreciate the honest and consistent references to prior works to help understand where the proof strategies were borrowed.
* The proofs and constructions are generally easy to follow, and the paper is overall well-written.
* There are also experiments to support the theoretical findings.
* The paper is the first to study generalized linear dueling bandits with Borda objective, to my knowledge. So, the results are not subsumed by any prior works.
* In particular, the first lower bounds are shown for this settin and a matching minimax upper bound is shown.

**Weaknesses:**

* I am curious why the authors pursued the Borda setting over the more established Condorcet setting where one targets a Condorcet winner and minimizes Condorcet regret. It is somewhat debatable in the literature which setting is preferable. In my opinion, since the Borda setting relies on pure exploration tactics (e.g., explore-then-commit in stochastic setting or EXP3 with T^{-1/3} exploration in adversarial setting), it is very amenable to the generalizing to this GLM model without hassle. Thus, I think the regret upper bounds in this paper are not surprising. The stochastic BETC-GLM regret bound seems to follow almost immediately from well-known sample complexity bounds for optimal design, where estimation of $\theta^*$ is completely decoupled from regret. Meanwhile, the adversarial regret bound for BEXP3 seems to be identical to that of DEXP3 in Saha et al., 2021 except for plugging in slightly different variance bounds at the end.
To contrast, in the Condorcet setting, where pure exploration is innapropriate to target $\sqrt{T}$ reget, one would have had to carefully decouple estimation of $\theta^*$ and regret minimization. So, I think the Condorcet setting would have been more technically interesting to study.
* Alternatively, it would have been more interesting to study instance-dependent regret rates (e.g., those appearing in Jamieson et al., 2015) as it's more unclear to me how those would behave for GLM dueling bandits.
* The adversarial regret upper bound seems to only be able to get the $(d\log(K))^{1/3}$ dependence and not the $d^{2/3}$ dependence if $K \gg 2^d$ because there is an unavoidable $\log(K)$ appearing in the EXP3 analysis. Therefore, it is not necessarily optimal in all regimes.
* As is the case for generalized linear MAB, there is a mysterious dependence on $\kappa^{-1}$ in the regret upper bounds. It is unclear to me if this dependence is optimal and calls into question how realistic this regret bound can be.

**Questions:**

# Questions
* As mentioned above, can the adverserial dueling bandit analysis be improved to $d^{2/3} T^{2/3}$ for very large $K$?
* Can the authors comment on the dependence in the regret of $\kappa^{-1}$ in the regret and whether it is optimal or realistic for common link functions?
* It seems like BEXP3 seems to perform the best in your experiments. This seems a little confusing to me because the constructed environments seem to be stochastic and not adversarial. Can the authors comment on this?

# Writing Notes
* The term "contextual vector" or "contextual dueling bandit" is used many times to refer to the feature ${\bf x}_i$ of arm $i$. This can be easily confused with contextual bandits where one observed a context $X_t$ independent of the arms, and so some clarification in the language might be helpful.

---

> ### Author Response · Authors · 2023-11-17
> **Author Response to Reviewer Kuh2**
>
> Thank you for your feedback and your suggestions on writing! We address your comments and questions as follows:
>
> ---
> **Q1**: Can the adversarial dueling bandit analysis be improved to $d^{2/3} T^{2/3}$  for very large K? \
> **A1**: Yes. The main idea is to use an $\epsilon$-covering argument. In the $d$-dimensional space, it suffices to choose $ O((1/\epsilon)^{d})$ representative vectors to cover all $K$ average contextual vectors $\frac{1}{K} \sum_{j=1}^{K}\phi_{i,j}$ with error up to $\epsilon$. Now BEXP3 will be performed as the original case except that at Line 7 in Alg 2, we replace the average contextual vectors $\frac{1}{K} \sum_{j=1}^{K}\phi_{i,j}$ with those nearest representative vectors. Since there are only $O((1/\epsilon)^{d})$ uniquely different rewards, the EXP3 argument (equation D.3) can be performed on $O((1/\epsilon)^{d})$ unique arms and eventually the algorithm suffers a regret of $\tilde{O}(d^{2/3} T^{2/3} \log^{1/3}(1/\epsilon))$. The $\epsilon$-net incurs an additional approximation error of order $O(\epsilon T)$. Setting $\epsilon = T^{-1}$ will improve the regret to $d^{2/3} T^{2/3}$ up to log factors. We add a section explaining this in detail (Appendix H) in the revised version.
>
> ---
>
>
> **Q2**: Can the authors comment on the dependence in the regret of  $\kappa^{-1}$ in the regret and whether it is optimal or realistic for common link functions? \
> **A2**: For general link functions, all previous works on generalized linear models suffer the same order of dependence on $\kappa^{-1}$ in their regret upper bounds. For some specific link functions, such as the logistic link function, [1] showed better dependence on $\kappa^{-1}$. Still, it is so far unknown to the community if $\kappa^{-1}$ for general link functions can be improved.  We will add comments on this in the revised version.
>
> ---
> **Q3**: BEXP3 seems to perform the best in your experiments. This seems a little confusing to me because the constructed environments seem to be stochastic and not adversarial. Can the authors comment on this?\
> **A3**: In principle, we don’t think it is surprising, because the adversarial setting strictly covers the stationary setting when the link function is linear. So BEXP3 should work at least as well as BETC, which is indicated by the regret upper bound. We believe this gap can be eliminated by carefully tuning the constant of each input parameter. Please see Appendix I for the additional experiment, where we show BETC performs better than BEXP3.
>
> ---
> \
> \
> [1] Improved Optimistic Algorithms for Logistic Bandits, Faury et al., ICML 2020

---

> > ### Comment · Reviewer_Kuh2 · 2023-11-19
> >
> > Thank you for the response and for addressing the question about optimal dependence on $d$. Overall, I would still say the novelty is limited due to the results being close adaptations of the corresponding results in dueling bandits with finite number of arms, especially in this Borda version of the problem. This point also seems to be echoed by some of the other reviewers. I will keep my score the same.

---

> > > ### Author Response · Authors · 2023-11-22
> > >
> > > Thanks for your further feedback.
> > >
> > > We are still looking forward to resolving your concerns on the novelty. As we have explained in our discussion with Reviewer vgjv, our problem setting is natural, and we chose the Borda winner as the learning objective simply because it is a meaningful objective that always exists even when there is no coherent ranking.
> > >
> > > As for your concerns regarding our contribution compared with the results in dueling bandits with finite number of arms [Saha et al 21], please allow us to explain the difference as follows:
> > >
> > > In the previous work [Saha et al 21], there are $K-1$ good arms and only the best one of them differs from the others. This design will not work in our setting as it leads to lower bounds sub-optimal in dimension $d$. The new construction of our lower bound is based on the hardness of identifying the best arm in the $d$-dimensional linear bandit setting, which is quite different from the multi-armed setting.
> > >
> > > [Saha et al 21]’s proof is directly based on hypothesis testing: either identifying the best arm with gap $\epsilon$ within $T$ rounds (if $T > \frac{K}{1440 \epsilon^3}$) or incurring $\epsilon T$ regret (if $T \le \frac{K}{1440 \epsilon^3}$). In contrast, our proof technique bounds from below the regret by the expected number of sub-optimal arm pulls and does not divide the problem instances into two cases (i.e. whether $T≤\frac{K}{1440 \epsilon^3}$).
> > > To prove the lower bound, we first apply a new reduction step to restrict the choice of $i_t$. Then we bound from below the regret by the expected number of sub-optimal arm pulls.

---

### Official Review · Reviewer_vgjv · 2023-11-04

**Soundness:** 2 fair
**Presentation:** 3 good
**Contribution:** 3 good
**Rating:** 5
**Confidence:** 3

**Summary:**

The authors prove a regret lower and upper bounds for the Borda regret minimization problem for generalized linear models with K arms, which largely depend polynomial on d is the dimension of the contextual vectors and T is the time horizon. Specifically, they achieve matching upper and lower bounds of d^{2/3}T^{2/3} for the stochastic setting, as well as a d^{1/3}T^{2/3} upper bound in the adversarial setting.

**Strengths:**

The authors extend the previous works by allowing the regret bound to not inherently depend the number of arms K (generally log(K)); rather it depends on the inherent dimensionality of contextual vectors, which are given apriori. They study both the adversarial and stochastic settings, with generally similar conclusions. Furthermore, their lower bounds demonstrate that their upper bounds are in fact tight due to the dual-regret nature of Borda regret. Their lower bounds seem to imply that the preference information + regret structure makes Borda regret minimization inherently harder than typical bandit regret settings as note that the action pair with the highest reward does not lead to optimal minimization.

**Weaknesses:**

The main weakness is the novelty of the paper and it's derived bounds. It appears that the lower bounds use the standard hypercube + info theoretical argument from [Dani et al 08 or survey on bandits by Lattimore] and fails to clarify the novelty in their lower bounds from previous works. Furthermore, the upper bound uses a simple ETC algorithm and analysis and it is unclear how the novelty from the typical ETC analysis [see survey on bandits by Lattimore].

Furthermore, the paper mentions Borda regret in the adversarial setting but it becomes less obvious why Borda regret in this setting is even possible without assumptions 3.2 and 3.3 (it appears that Algorithm 2 does not use the structure of mu at ALL!). If that is indeed possible, why is there no reduction from adversarial to stochastic?

**Questions:**

In algorithm 2 (adversarial setting), where does mu show up? How can it work without inferring mu at all and without assumptions 3.2/3.3?

Can you explain the novelty in your lower/upper bounds in the stochastic setting?

---

> ### Author Response · Authors · 2023-11-17
> **Author Response to Reviewer vgjv**
>
> Thank you for your feedback! We address your comments and questions as follows:
>
> ---
> **Q1**: In algorithm 2 (adversarial setting), where does $\mu$ show up? How can it work without inferring $\mu$ at all and without assumptions 3.2/3.3? \
> **A1**: We would like to clarify that in the adversarial setting, we consider linear models rather than generalized linear models. In detail, we consider the linear link function is $\mu(x) = \frac{1}{2} + x$ in the adversarial setting, and it satisfies Assumption 3.2 with $\mu’(\cdot) = 1$ and Assumption 3.3 with $L_{\mu}=M_{\mu}=1$.
>
> We will add comments on this in the revised version to avoid any misunderstanding.
>
> ---
> **Q2**: Can you explain the novelty in your lower/upper bounds in the stochastic setting? \
> **A2**: One key observation in solving the stochastic setting is that, to estimate the Borda score, the most sample-efficient way is to query each pair uniformly. In the linear setting, this means we need to explore each direction in $\mathbb{R}^d$ uniformly well.
>
> Converting this idea into the regret-minimization setting requires new techniques. For the lower bound, this leads to our design of good/bad arms. In the previous work [Saha et al 21], there are $K-1$ identical arms against one best arm. This design will not work in our setting as it leads to lower bounds sub-optimal in dimension $d$. The new construction of our lower bound is based on the hardness of identifying the best arm in the $d$-dimensional linear bandit model.  To prove the lower bound, we first apply a new reduction step to restrict the choice of $i_t$. Then we bound from below the regret by the expected number of sub-optimal arm pulls.
>
> For the upper bound, our hard instance construction already sheds light on the algorithm design (see comments below Theorem 4.1):  to differentiate the best item from its close competitors, the algorithm must query the bad items to gain information. This means BETC algorithms should naturally be optimal and do not need further complication. To explore each direction in $\mathbb{R}^d$ uniformly, we adopt the G-optimal design to explore all directions efficiently.

---

> > ### Comment · Reviewer_vgjv · 2023-11-17
> >
> > Thanks for the clarifications! Generally, it would still seem that your work seems to be a combination of [Saha et al 21] and standard (generalized) linear bandit upper/lower bounds. Therefore, I would be hesitant to say that the novelty is sufficient for a clear accept.
> >
> > My main concern with a marginal accept is that in this model, the reward received as feedback does not clearly correspond to the regret incurred in the round. This allows for generally easier lower bounds and I am unsure why the Borda regret model provides interesting theoretical or empirical insights.

---

> > > ### Author Response · Authors · 2023-11-22
> > >
> > > Thanks for your further feedback.
> > >
> > > Could you please further clarify "the reward received as feedback does not clearly correspond to the regret incurred in the round"? In our setting, the feedback is the noisy comparison between two items, which is the same as any work regarding dueling bandits. The goal is to identify the Borda winner, and in the context of regret minimization, it is natural to define the gap between the Borda scores as the regret, because a sub-linear regret indicates the algorithm will eventually find the (approximate) optimal Borda winner.
> > >
> > > It seems that you are also concerned that the Borda winner is "easier" or not interesting enough. As discussed in the paper, we do not assume a coherent ranking, and learning under this scenario requires some objective.
> > > In this paper, we choose the learning objective to be the Borda winner that always exists even without a coherent ranking.
> > > We are happy to discuss other potential options such as the Copeland winner or the von Neumann winner, but there is no evidence showing the Borda winner is the “easier” one.

---

### Author Response · Authors · 2023-11-17
**Thank you**

Dear reviewers,

We have revised the paper according to your suggestions. The revised part is highlighted in blue color. We mainly address the novelty, the improved upper bound for BEXP3 with the covering number argument (Appendix H), and an updated experiment figure that shows the similar performance of BETC and BEXP3 (Appendix I).

Thank you for all the suggestions!
Authors

---

### Meta-Review · Area_Chair_a8Mp · 2023-12-04

**Metareview:**

This paper combines GLM bandits with dueling bandits and a concept of Borda winner (to measure the regret). The lower bound are obtained using rather standard techniques and the algorithm is some non-adaptive version of ETC (that therefore reaches a T^{2/3} regret)


The reviewers raised several concerns with this paper, the more important one being the relative marginal contribution to the field, both conceptually and technically.

I feel that they all found this paper interesting, and technically sound, but without being too thrilled about it. I Agree with them, this is a nice paper, but not good enough to reach the ICLR bar.

**Justification For Why Not Higher Score:**

It does not reach the bar

**Justification For Why Not Lower Score:**

N/A

---

### Decision · Program_Chairs · 2024-01-16

Reject